# MoEQuant: Enhancing Quantization for Mixture-of-Experts Large Language Models via Expert-Balanced Sampling and Affinity Guidance

Zhixuan Chen [* 1]  Xing Hu [* 1]  Dawei Yang[✉ * 1]
Zukang Xu [1]  Chen Xu [1]  Zhihang Yuan [1]  Sifan Zhou [1 2]  Jiangyong Yu [1]

## Abstract

Mixture-of-Experts (MoE) large language models (LLMs), which leverage dynamic routing and sparse activation to enhance efficiency and scalability, have achieved higher performance while reducing computational costs. However, these models face significant memory overheads, limiting their practical deployment and broader adoption. Post-training quantization (PTQ), a widely used method for compressing LLMs, encounters severe accuracy degradation and diminished generalization performance when applied to MoE models. This paper investigates the impact of MoE's sparse and dynamic characteristics on quantization and identifies two primary challenges: (1) **Inter-expert imbalance**, referring to the uneven distribution of samples across experts, which leads to insufficient and biased calibration for less frequently utilized experts; (2) **Intra-expert imbalance**, arising from MoE's unique aggregation mechanism, which leads to varying degrees of correlation between different samples and their assigned experts. To address these challenges, we propose MoEQuant, a novel quantization framework tailored for MoE LLMs. MoEQuant includes two novel techniques: 1) **Expert-Balanced Self-Sampling (EBSS)** is an efficient sampling method that efficiently constructs a calibration set with balanced expert distributions by leveraging the cumulative probabilities of tokens and expert balance metrics as guiding factors. 2) **Affinity-Guided Quantization (AGQ)**, which incorporates affinities between experts and samples into the quantization process, thereby accurately assessing the impact of individual samples on different experts within the MoE layer. Experiments demonstrate that MoEQuant achieves substantial

performance gains (more than 10 points accuracy gain in the HumanEval for DeepSeekMoE-16B under 4-bit quantization) and boosts efficiency.

## 1. Introduction

Recent advances in natural language processing have been profoundly influenced by the success of large language models (LLMs). Among these, Mixture-of-Experts (MoE) LLMs, which leverage the dynamic routing mechanisms and scalable capabilities of MoE layers, have demonstrated superior performance and achieved state-of-the-art results, garnering significant attention from the research community (Jiang et al., 2024; Qwen, 2024; Liu et al., 2024b). However, during deployment, MoE LLMs face not only the same memory bandwidth constraints as conventional LLMs (Kim et al., 2023; Dettmers et al., 2022) but also substantially higher storage requirements. For example, in Qwen-MoE-A2.7B-14B (Qwen, 2024), only 2.7 billion parameters are activated during the generation phase, yet all 14 billion parameters must reside in memory, significantly increasing inference costs. Furthermore, MoE layers account for most of the parameter footprint within the transformer blocks: approximately 80% when considering activated experts and up to 97% when including all experts. Consequently, compressing MoE LLMs, particularly their MoE layers, is critical for reducing inference costs and enabling deployment on resource-constrained devices with limited memory capacity and bandwidth.

Post-Training Quantization (PTQ), which quantizes weights into a low-precision format, effectively reduces model size and memory footprint, achieving notable success in conventional large language models (LLMs). For example, AWQ (Lin et al., 2023) and GPTQ (Frantar et al., 2022) compress model weights by up to four times without requiring additional training, while maintaining nearly lossless performance. However, when these methods are applied directly to MoE LLMs, they often lead to overfitting and significant performance degradation, particularly in terms of generalization. This is because they focus on layer-wise quantization while overlooking the unique architecture of MoE, which routes samples to a limited number of experts and aggregates their outputs through weighted combinations.

---

[*]Equal contribution  [1]Houmo AI [2]Southeast University. Correspondence to: Dawei Yang <dawei.yang@houmo.ai>.

*Proceedings of the 42nd International Conference on Machine Learning*, Vancouver, Canada. PMLR 267, 2025. Copyright 2025 by the author(s).

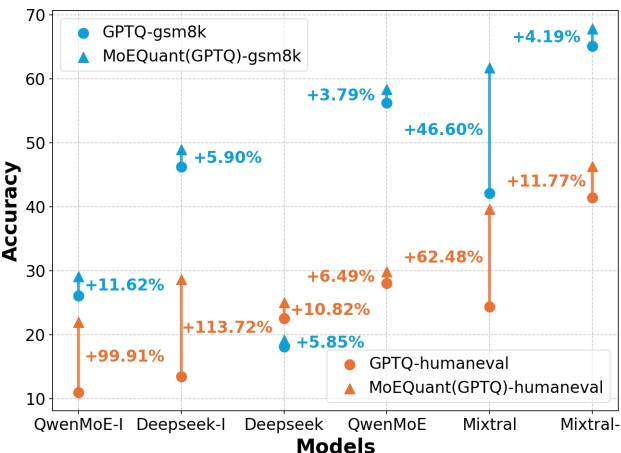

*Figure 1.* The relative accuracy gains of GPTQ across various models for two generative tasks, HumanEval and GSM8k, before and after applying MoEQuant are presented. The suffix "I" denotes the instruction fine-tuned version.

In addition, they fail to account for the inherent sparsity and heterogeneity introduced by the MoE structure.

We perform a comprehensive analysis of the key factors that affect the quantization performance of MoE LLMs and identified two inherent imbalances within the MoE architecture as the primary contributors. **Firstly,** there is an imbalance in the distribution of samples across different experts. As highlighted in DeepSeek (Liu et al., 2024b), various techniques have been developed to maintain load balance among experts, which is equally critical during the calibration phase. However, calibration sets are often domain specific, and the gating mechanism can result in some experts being overloaded while others remain underloaded. Underloaded experts naturally receive insufficient calibration, leading to significant quantization errors. As shown in Figure 2, both of the two most commonly used calibration sets exhibit this imbalance. **Secondly,** there is an imbalance in the affinities between samples and their assigned experts. Unlike traditional LLMs, where all samples are processed by a single feedforward network, MoE architectures use a gating mechanism to express the output as a weighted sum of results from multiple experts. Consequently, from the perspective of each expert, samples exhibit varying levels of affinity, defined as the correlation between a sample and its assigned expert. Existing PTQ methods (Xiao et al., 2022; Lin et al., 2023; Ashkboos et al., 2024) fail to account for this affinity during expert quantization. For example, GPTQ (Frantar et al., 2022) disregards the impact of the gating unit when collecting Hessian information, resulting in an inaccurate assessment of the importance of individual samples for each expert. This oversight distorts the Hessian information and significantly degrades the performance of the quantized model.

To address the aforementioned two imbalances, this paper introduces two methods: Expert-Balanced Self-Sampling

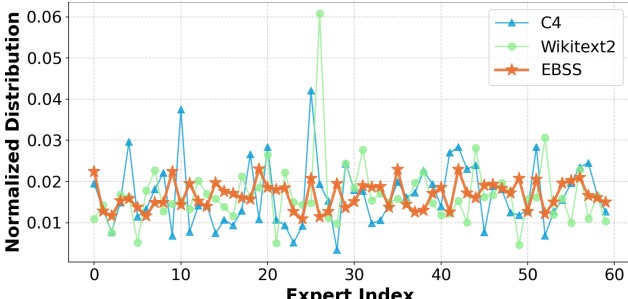

*Figure 2.* Sample distribution on the first MoE layer of Qwen-MoE-A2.7B-14B for different calibration sets. For C4 and WikiText2, 128 × 512 tokens were sampled, for our EBSS, samples were generated through the model's self-sampling method.

(EBSS) and Affinity-Guided Quantization (AGQ). **EBSS** constructs calibration sets based on the self-sampling capabilities of LLMs and incorporates cumulative probability and expert balance metrics to guide the sampling process. This guidance significantly reduces search complexity. Additionally, it ensures that calibration samples are evenly distributed among experts and consistent with the pretraining data distribution. **AGQ** addresses the imbalance in token-expert affinity during expert quantization by integrating affinity into layer-wise calibration and constructing weighted quantization errors. This approach adapts to the dynamic characteristics of MoE, enabling more accurate calculation of quantization errors and sensitivity. By integrating these two methods, we present MoEQuant, a framework that bridges existing quantization techniques with MoE architectures, taking a crucial step toward reconciling the efficiency of quantized systems with the unique requirements of MoE LLMs. As shown in Figure 1, MoEQuant achieves performance improvements of varying degrees across different models, highlighting its effectiveness and broad applicability for enhancing MoE language models. Our contributions are summarized as follows:

- We identify two critical imbalances—inter-expert and intra-expert—in the quantization of MoE models: sample distribution imbalance among experts and token-expert affinity imbalance.

- We propose Expert-Balanced Self-Sampling to efficiently generate a balanced calibration dataset, ensuring equitable utilization of all experts. We also propose Affinity-Guided Quantization to introduce token-expert affinities into the quantization process, thereby improving weight update accuracy and reducing quantization errors.

- We develop MoEQuant, which seamlessly integrates EBSS and AGQ with existing PTQ methods, significantly enhancing the quantization performance of MoE LLMs. As one of the first studies in this area, we will release the code to encourage further exploration and drive progress in this field.

## 2. Related Work

### 2.1. Mixture-of-Experts Large Language Models

The Mixture-of-Experts (MoE) model, first introduced by (Jacobs et al., 1991) and (Jordan & Jacobs, 1994), has been extensively explored in the various contexts (Eigen et al., 2013; Theis & Bethge, 2015; Deisenroth & Ng, 2015; Aljundi et al., 2017). In MoE LLMs, each MoE layer comprises multiple expert networks and a gating network. The gating network, typically implemented as a linear layer with a softmax function, directs inputs to the appropriate expert networks and aggregates their outputs. Different models employ various configurations. For example, Switch-Transformer (Fedus et al., 2022) introduces a top-1 gating strategy, achieving competitive results for specific model sizes. Mixtral-8x7B (Jiang et al., 2024) combines MoE with infrastructure innovations, utilizing two experts per layer to achieve excellent performance while maintaining low computational cost. DeepSeekMoE (Dai et al., 2024) refines expert segmentation by subdividing the intermediate hidden dimensions of FFNs, increasing the number of experts, and activating more of them to improve knowledge decomposition and capture. It also introduces shared experts, which are always activated to consolidate common knowledge across contexts, reducing parameter redundancy in routing-specific experts. DeepSeekv2 (Liu et al., 2024a) and DeepSeekv3 (Lu, 2025) further enhance performance with refined designs. Qwen-Moe (Qwen, 2024) replaces traditional FFN layers entirely with MoE layers, employing four shared experts alongside four unshared experts selected from a pool of 60. During training, Qwen-MoE first adapts the existing Qwen-1.8B model to create Qwen1.5-MoE-A2.7B-16B, achieving better overall pretraining performance.

### 2.2. Post-Training Quantization for LLMs

Most LLMs are built upon Transformer(Vaswani et al., 2017) architecture, which is inherently memory-intensive. Post-training quantization (PTQ) has become a widely adopted approach to compress LLMs, effectively reducing memory consumption while maintaining model accuracy. Two prominent PTQ methods, GPTQ (Frantar et al., 2022) and AWQ (Lin et al., 2023), have been extensively studied. GPTQ employs Hessian-based error compensation to minimize quantization errors and achieve high compression rates. AWQ, on the other hand, accounts for the impact of activation distributions on weight quantization, thereby improving the performance of quantization. Beyond these methods, several advanced techniques have emerged to further enhance PTQ. Quarot (Ashkboos et al., 2024) applies Hadamard transformations to remove outliers without altering the output, thus enhancing the effectiveness of GPTQ. GPTVQ (van Baalen et al., 2024) explores non-uniform

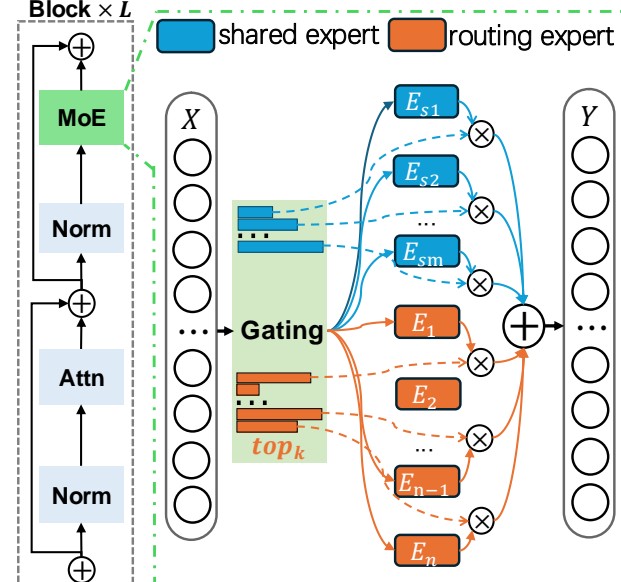

*Figure 3.* The MoE structure in LLMs. The router selects all non-shared experts and $k$ shared experts with highest confidence. The predictions from all experts are then aggregated and weighted.

quantization schemes from a vector perspective, offering better adaptability to weight distributions.

However, these methods overlook the unique challenges posed by MoE architectures, resulting in significant accuracy drop. Our proposed MoEQuant, rooted in the relationship between calibration samples and experts, is orthogonal to existing PTQ methods, enabling seamless integration for the effective quantization of MoE-based LLMs.

## 3. Preliminaries

As illustrated in Figure 3, an MoE layer comprises $m$ shared and $n$ routing experts, along with a gating network that assigns different experts and their corresponding probabilities to each token. Only the top $k$ shared experts with the highest affinities are utilized. For a given input token $\boldsymbol{x}$, the output $\boldsymbol{y}$ is computed as a weighted sum of the outputs from the top $k$ routing experts and all shared expert:

$$y = \underbrace{\sum_{i=1}^{m} E_i^s(\boldsymbol{x})g_i(\boldsymbol{x})}_{\text{shared experts}} + \underbrace{\sum_{j \in \mathcal{K}} E_j^r(\boldsymbol{x})g_j(\boldsymbol{x})}_{\text{top } k \text{ routing experts}}, \quad (1)$$

where $\mathcal{K} = \text{top}k\left(\{g_i(\boldsymbol{x}) \mid i \in \{1, \dots, m\}\}\right)$, $g_i(\boldsymbol{x})$ represents the weight assigned to the $i$-th expert.

**Perplexity (PPL)** is a common metric for evaluating the quality of language models. A lower PPL indicates better predictive accuracy and closer alignment with the model's true distribution. For a sequence composed of $n$ tokens $\mathcal{D} = (d_1, d_2, \dots d_n)$, the perplexity with respect to model

$\mathcal{M}$ is defined as:

$$\text{PPL}(\mathcal{D} \mid \mathcal{M}) = \exp\left(-\frac{1}{N}\sum_{i=1}^{N}\log P_{\mathcal{M}}\left(d_i \mid d_1, d_2, \ldots, d_{i-1}\right)\right) \tag{2}$$

where $P\left(d_i \mid d_1, d_2, \ldots, d_{i-1}\right)$ represents the probability predicted by the model for $d_i$, given the context $d_1, d_2, \ldots, d_{i-1}$.

**Expert balance** is evaluated by the standard deviation in the frequency of expert usage across all layers. We define it as:

$$\sigma = \frac{\sum_{l=1}^{L}\sigma_l}{L} \tag{3}$$

$$\sigma_l = \sqrt{\frac{1}{E-1}\sum_{e=1}^{E}(u_l^e - \hat{u}_l)^2} \tag{4}$$

where $L$ denotes the number of layers in the MoE model, $E$ represents the total number of experts in a layer, and $\sigma_l$ refers to the standard deviation of the $l$-th layer, which is calculated based on the usage frequency $u_l^e$ of each expert and the average frequency $\hat{u}_l$.

**Quantization** typically involves mapping a floating-point number to a discrete interval using integer values. For weight quantization, we focus on the most commonly used per-channel symmetric uniform quantization. The quantization process is expressed as follows:

$$\mathcal{Q}(\boldsymbol{W}) = \text{clamp}\left(\left\lfloor\frac{\boldsymbol{W}}{\boldsymbol{s}}\right\rceil, q_{\min}, q_{\max}\right), \tag{5}$$

where $\boldsymbol{W} \in \mathbb{R}^{o \times c}$ represents the weight matrix, $\boldsymbol{s} \in \mathbb{R}^{o}$ denotes the channel-wise quantization step, and $q_{min}, q_{max}$ represent the quantization bounds. To facilitate the evaluation of quantization error, we typically perform a dequantization operation:

$$\hat{\boldsymbol{W}} = \mathcal{Q}(\boldsymbol{W}) \cdot \boldsymbol{s} \tag{6}$$

For a linear layer, the loss caused by quantizing $\boldsymbol{W}$ can be formulated as

$$\mathcal{L}(\hat{\boldsymbol{W}}) = \left\|\boldsymbol{W}\boldsymbol{X} - \hat{\boldsymbol{W}}\boldsymbol{X}\right\|_F^2, \tag{7}$$

where $\boldsymbol{X} \in \mathbb{R}^{b \times c}$ represents the activation of the calibration data at this layer. AWQ (Lin et al., 2023) utilizes Equation 7 to guide the selection of smoothing coefficients and weight pruning. GPTQ (Frantar et al., 2022) follows OBQ (LeCun et al., 1989), which uses the Hessian to compensate for the quantization error. In conjunction with Equation 7, the Hessian can be effectively computed as:

$$\boldsymbol{H} = \boldsymbol{X}\boldsymbol{X}^\top \tag{8}$$

# 4. Method

## 4.1. MoEQuant

In this paper, we present MoEQuant, a framework designed to efficiently quantize LLMs utilizing MoE architectures. MoEQuant addresses the critical challenge of expert imbalance, both inter- and intra-expert, which arises during the quantization process. MoEQuant tackles the imbalance from two perspectives: the generation of expert-balanced calibration datasets and the token-expert correlation during expert calibration. Correspondingly, it incorporates two solutions: Expert-Balanced Self-Sampling (EBSS) and Affinity-Guided Quantization (AGQ). EBSS generates calibration samples that ensure the balanced engagement of all experts within MoE architectures. AGQ, on the other hand, addresses the correlation disparities between samples introduced by gating units in MoE layers.

Both methods are plug-and-play and can be seamlessly integrated with other quantization techniques to improve the performance of MoE LLMs. Detailed descriptions of them are provided in Sections 4.2 and 4.3.

## 4.2. Expert-Balanced Self-Sampling

Current PTQ methods typically rely on domain-specific calibration datasets, such as WikiText-2. Although these calibration datasets can preserve reasonable generalization capabilities for standard LLMs, their direct application to MoE LLMs often leads to significant performance degradation. This degradation occurs because domain-specific calibration datasets result in an uneven sample distribution among experts. As illustrated in Figure 2, relying on a single calibration set usually produces a long-tailed distribution of samples among different experts.

An intuitive approach is to construct a domain-balanced calibration set by sampling data from multiple domains. However, the virtually infinite number of possible domains makes achieving true domain balance both complex and impractical. Moreover, as shown in Figure 4, even high-quality datasets often exhibit high perplexity, indicating a misalignment between the selected data and the model's inherent distribution.

**Problem Definition** Based on the above, the objective is to identify a dataset $\mathcal{D}^*$ that satisfies two key properties:

- Low perplexity. The samples in $\mathcal{D}^*$ should align closely with the inherent distribution of the model $\mathcal{M}$, which corresponds to minimizing perplexity.

- Expert balance. The samples should be evenly distributed among experts in MoE LLMs, ensuring that no expert is overused or underused.

This dual requirement can be formulated as a joint optimiza-

tion problem, which can be formulated as:

$$\mathcal{D}^* = \arg\min_{\mathcal{D}} \left\{ \text{PPL}(\mathcal{M}, \mathcal{D}) \cdot \exp\left(\frac{\sigma(\mathcal{M}, \mathcal{D})}{\tau}\right) \right\}, \quad (9)$$

where $\exp\left(\frac{\sigma(\mathcal{M},\mathcal{D})}{\tau}\right)$ represents the reciprocal of normalized imbalance, $\exp$ is used to normalize $\sigma$, and $\tau$ is a hyper-parameter controlling the impact of expert imbalance. Regarding the influence of the hyperparameter $\tau$ and the objective function, we provide a detailed derivation of the complete process in the Appendix A.1.

The perplexity optimization corresponds to an optimal subset selection problem, while expert balancing is analogous to a load-balancing problem. Both are NP-hard, making a direct solution computationally infeasible. To enable practical analysis, the problem can be reformulated in combination with Equation 2 as

$$\mathcal{D}^* = \arg\min_{\mathcal{D}} \left\{ \frac{-1}{N} \sum_{i=1}^{n} \left(\log\left(P(\mathcal{D}_i|\mathcal{D}_{1:i-1})\right)\right) + \frac{\sigma(\mathcal{M}, \mathcal{D})}{\tau} \right\},$$

$$\text{subject to } \mathcal{D} \in = \underbrace{\mathcal{V} \otimes \mathcal{V} \otimes \cdots \otimes \mathcal{V}}_{n \text{ times}},$$

$$(10)$$

where $\mathcal{V} = \{v_1, v_2, \ldots, v_m\}$ represents the vocabulary that contains $m$ tokens, $P$ denotes probabilities predicted by $\mathcal{M}$. $n$ is the sequence length and $\otimes$ denotes the Cartesian product. In this context, optimization can be viewed as searching for the optimal path within an $n$-dimensional vocabulary space.

**Challenges.** One challenge lies in the availability of limited datasets. Since only a small amount of data is typically accessible for calibration, it is difficult to ensure ideal domain balance or alignment with the pre-training distribution, which can adversely affect the final quantization performance. Another challenge is the computational cost of searching through the vast space of potential calibration sets. A brute-force search would require exploring $m^n$ possibilities, which is infeasible. Greedy search strategies, although more efficient, may suffer from local optima, highlighting the need for more sophisticated but efficient search methods.

**Self-Sampling.** To address the challenge of data availability, we leverage the self-sampling capabilities of LLMs to construct calibration data. This data-free approach relies solely on the model's vocabulary and is naturally consistent with the model's learned language distribution (Liu et al., 2023). Furthermore, during the self-sampling process, historical probabilities and expert distributions are cached, eliminating redundant computations for perplexity and expert balance metrics. We define the historical cumulative log-probability of a sequence $S$ as:

$$R_S = \sum_{i=1}^{n} \log(P(S_i|S_{1:i-1}), \quad (11)$$

where $n$ is the length of $S$. During the sampling phase, perplexity can be easily calculated by $R_S$ and the predicted probability $P(v|S)$ by

$$\text{PPL}(\mathcal{M}, S\|v) = \exp\left(\frac{-1}{n+1}(R_s + P(v|S))\right) \quad (12)$$

where $\|$ denotes concatenation.

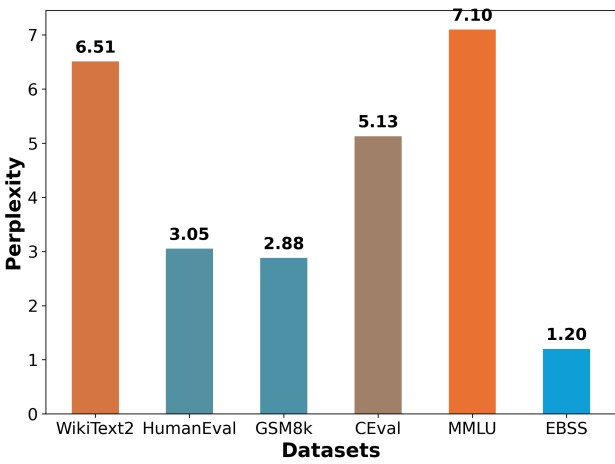

Figure 4. Perplexity performance on DeepSeek-MoE-16B of different datasets.

**Probability-Guided Path Pruning.** When predicting the next token in a self-sampling approach, the head layer outputs probabilities for all possible candidates, which typically exhibit a multimodal distribution. Tokens with low probabilities often result in incoherent or semantically incorrect sequences. Based on this observation, we propose a probability-guided path pruning method to effectively improve search efficiency. The core idea is to ignore low-probability branches during the search process.

Specifically, during the calibration dataset search, we retain only $w$ branches $\mathcal{S} = \{S^1, S^2, ..., S^w\}$, each with a length of $l$. When generating candidate sequences of length $l+1$, each branch $S^t$ expands to potential sequences within the space $S^t \otimes V$. The pruning evaluation metric for these sequences is defined as:

$$\text{score}(S\|v) = \frac{-1}{l+1}(R_S + \log P(v|S)) + \frac{\sigma(\mathcal{M}, S)}{\tau},$$

$$\text{subject to } v \in V,$$

$$(13)$$

where $S$ is one of $\mathcal{S}$, and $R_S$ corresponds to the cumulative probability defined in Equation 11. The process of generating a new set of $w$ candidate sequences, $\hat{S} = \{\hat{S}^1, \hat{S_s}^1, ..., \hat{S}^w, \}$, is expressed as:

$$\hat{\mathcal{S}} = \arg\operatorname*{topk}_{S\|v} (w, \text{score}(S\|v)),$$

$$\text{subject to } v \in V \text{ and } S \in \{S^i, S^2, ..., S^w\}, \quad (14)$$

where $\arg \operatorname{topk}_x (f(w,x))$ denotes the set of $w$ values of $x$ that maximize $f(x)$.

As indicated in Equation 14 and Figure 5, the scores of candidate sequences generated from the same input sequence $S$ are influenced by the probability distribution produced by the LLM. Additionally, the expert balance metric affects all candidate sequences derived from $S$. By introducing an efficient search method for the calibration set, the search complexity is significantly reduced from $O(m^n)$ to $O(wn)$, and the risk of local optima is effectively mitigated.

**Deferred Expert Imbalance Calculation**. It is important to note that during the pruning process, as shown in Equation 14, the evaluation metric does not incorporate the candidate token $v$ in the assessment of expert balance. This approach is justified for several reasons:

- Unlike perplexity, which can be directly computed from the probability distributions output by LLMs, calculating the expert distribution for each token in the vocabulary requires iterating over all possible tokens, a computationally expensive process. Since the distribution of the current sequence is already known, deferred computation incurs minimal additional cost.
- The pruning process relies primarily on the predicted probabilities of the LLM rather than on expert balance. Including expert distributions during pruning of the next token is inappropriate, as it may lead to semantic misalignment or incoherence.
- As demonstrated in Equation 14, the deferred calculation actually performs branch-level pruning, thereby ensuring the creation of an expert-balanced calibration set on the premise of maintaining the perplexity.

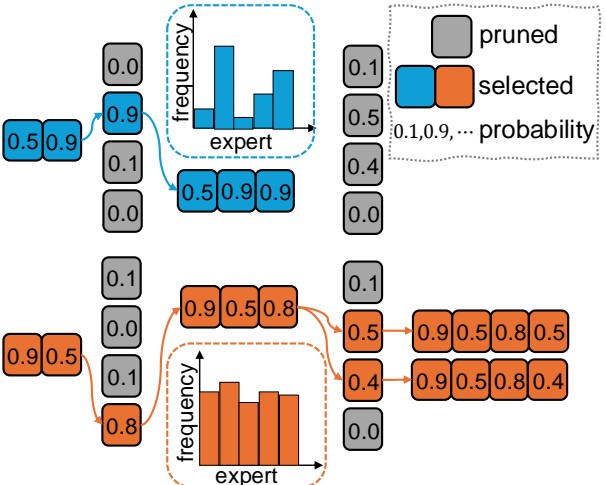

*Figure 5.* Illustrative diagram of EBSS. The expert distribution and cumulative probabilities jointly guide the path searching.

### 4.3. Affinity-Guided Quantization

In an MoE layer, the gating network assigns a probability score to each expert based on the input. The most relevant experts are selected and their outputs are weighted according to their assigned probabilities. Traditional layerwise quantization employs Equation 7 to uniformly minimize the quantization error but overlooks the probabilities between samples and experts. We define this correction as affinity, asserting that it is equally important and should be taken into account during the quantization process.

Let $E$ be a specific expert and denote the set of $b$ tokens routed to this expert as $\boldsymbol{X} = \{\boldsymbol{x_1}, \boldsymbol{x_2}, ..., \boldsymbol{x_b}\}$, with the corresponding affinity scores $\boldsymbol{c} = \{c_1, c_2, ..., c_b\}$ provided by the gating network. The output for the $i$-th token processed by this expert can be expressed as

$$\boldsymbol{y}_i = c_i E(\boldsymbol{x_i}). \tag{15}$$

$E$ is a FFN, which can be expanded into a sequence of linear layers and an activation function such as ReLU. Here, we consider a representative FFN structure, leading to:

$$\boldsymbol{y_i} = c_i \left\{ \left((\boldsymbol{x}\boldsymbol{W}^{up}) \odot f(\boldsymbol{x}\boldsymbol{W}^{gate})\right) \boldsymbol{W}^{down} \right\}, \tag{16}$$

where $f$ denotes the activation function, $\boldsymbol{W}$ denotes the parameter matrix. Because of the predominantly linear nature of the FFN and the quasi-linear property of $f$, the expression above can be reformulated as follows:

$$\boldsymbol{y_i} = \left((c_i\boldsymbol{x}\boldsymbol{W}^{up}) \odot f(\boldsymbol{x}\boldsymbol{W}^{gate})\right) \boldsymbol{W}^{down}$$
$$\boldsymbol{y_i} = \left((\boldsymbol{x}\boldsymbol{W}^{up}) \odot f(\boldsymbol{x}\boldsymbol{W}^{gate})\right) (c_i\boldsymbol{W}^{down}) \tag{17}$$
$$\boldsymbol{y_i} \approx \left((\boldsymbol{x}\boldsymbol{W}^{up}) \odot f(c_i\boldsymbol{x}\boldsymbol{W}^{gate})\right) \boldsymbol{W}^{down}$$

This shows that the token-expert affinity $c_i$ propagates through every layer of the expert network. When focusing on a specific linear layer, $c_i$ can be directly integrated into the layer's operations. In other words, different tokens exert a gate-aware influence on the weights of the same expert, with $c_i$ acting as a scaling factor that modulates the contribution of each token's input features to the linear layer.

**Affinity-aware quantization error.** Traditional quantization methods for LLMs have not taken into account the affinity-aware property. Here, we incorporate the gating coefficients into layer-wise calibration for the first time by redefining the quantization loss for $\boldsymbol{W}$ as

$$\mathcal{L}(\hat{\boldsymbol{W}}) = \sum_{i=0}^{n} c_i \cdot \left\| \boldsymbol{W}\boldsymbol{x}_i - \hat{\boldsymbol{W}}\boldsymbol{x}_i \right\|_F^2. \tag{18}$$

For PTQ methods based on quantization error, such as AWQ, Equation 18 incorporates token-expert affinity into the quantization process. Unlike the original implementation, which treats all tokens equally during calibration, our affinity-aware metric emphasizes tokens with higher affinities, thereby reducing the overall quantization error for influential tokens.

*Table 1.* Results of RTN, Omniquant, AWQ, GPTQ, Quarot+GPTQ and ours MoEQuant with 4-bit Weight Quantization among 9 Tasks on Qwen-MoE-14B, DeepSeekMoE-16B and Mixtral-8x7B. where + denotes MoEQuant based on AWQ, ++ denotes MoEQuant based on Quarot+GPTQ. Notably, except for our proposed MoEQuant, other methods utilize Wikitext2 as the calibration dataset, which leads to overfitting on Wikitext2. Perplexity measured on the C4 dataset more accurately reflects the performance of different methods.

| MODEL | METHOD | PPL | | ACCURACY | | | | | | | |
| | | WIKI TEXT2 | C4 | MMLU | HUMAN EVAL | GSM8K | BOOLQ | HELLA SWAG | OPEN BOOKQA | MATH QA | AVG. |
|---|---|---|---|---|---|---|---|---|---|---|---|
| QWEN-MoE-14B | FP | 7.22 | 9.30 | 59.60 | 32.32 | 62.55 | 79.82 | 57.96 | 30.40 | 35.77 | 51.20 |
| | RTN | 10.83 | 12.49 | 48.10 | 14.63 | 16.07 | 72.11 | 51.42 | 25.80 | 30.08 | 36.89 |
| | OMNIQUANT | 7.67 | 9.98 | 56.30 | 31.71 | 52.39 | 78.20 | 56.58 | 29.40 | 33.63 | 48.31 |
| | AWQ | 8.59 | 10.93 | 51.63 | 20.73 | 36.77 | 71.96 | 54.78 | 30.40 | 31.39 | 42.52 |
| | *MOEQuant*+ | 8.77 | 10.67 | 52.33 | 22.10 | 42.22 | 74.52 | 54.92 | **30.40** | 33.44 | 44.27 |
| | GPTQ | 8.00 | 10.99 | 53.70 | 20.73 | 22.82 | 73.52 | 52.70 | 29.40 | 28.27 | 40.16 |
| | QUAROT+GPTQ | 7.43 | 10.11 | 57.90 | 28.05 | 56.25 | **78.77** | 56.54 | 29.00 | **36.48** | 49.00 |
| | *MOEQuant*++ | 7.55 | 9.62 | **58.30** | **29.87** | 58.38 | 78.04 | **56.87** | 30.20 | 35.50 | **49.59** |
| DEEPSEEK-MoE-16B | FP | 6.51 | 9.04 | 44.60 | 26.83 | 20.16 | 72.72 | 58.06 | 32.20 | 31.49 | 40.86 |
| | RTN | 7.47 | 10.01 | 36.10 | 18.90 | 10.54 | 70.21 | 55.76 | 30.60 | 28.87 | 35.85 |
| | OMNIQUANT | 6.79 | 9.49 | 43.50 | 21.95 | 18.65 | 73.82 | 56.67 | 32.40 | 31.02 | 39.72 |
| | AWQ | 6.80 | 9.50 | 40.57 | 25.00 | 17.06 | 71.65 | 56.42 | 32.20 | 31.76 | 39.23 |
| | *MOEQuant*+ | 6.94 | 9.32 | 41.20 | 25.00 | 18.90 | 71.98 | 56.79 | **32.12** | **31.82** | 39.68 |
| | GPTQ | 6.82 | 10.35 | 39.60 | 21.34 | 11.60 | 72.14 | 56.05 | 30.60 | 30.35 | 37.38 |
| | QUAROT+GPTQ | 6.66 | 9.39 | 40.60 | 22.56 | 19.18 | 72.17 | 57.03 | 30.60 | 30.95 | 39.01 |
| | *MOEQuant*++ | 6.78 | 9.22 | **42.20** | **25.00** | **19.18** | **73.49** | **57.20** | 31.40 | 31.66 | **40.01** |
| MIXTRAL-8x7B | FP | 3.84 | 6.87 | 70.50 | 32.93 | 65.88 | 85.23 | 64.88 | 35.80 | 42.41 | 56.80 |
| | RTN | 5.41 | 8.13 | 62.20 | 28.05 | 27.90 | 80.85 | 61.73 | 32.20 | 37.35 | 47.18 |
| | OMNIQUANT | 4.19 | 7.20 | 68.10 | 34.75 | 57.01 | 84.13 | 63.03 | 33.00 | 41.91 | 54.56 |
| | AWQ | 5.01 | 7.98 | 62.75 | 25.00 | 38.67 | 79.97 | 62.11 | 33.60 | 38.43 | 48.64 |
| | *MOEQuant*+ | 5.15 | 7.84 | 64.66 | 25.45 | 50.66 | 81.03 | 62.73 | **34.00** | 39.77 | 51.19 |
| | GPTQ | 4.84 | 8.08 | 64.30 | 24.39 | 42.15 | 83.03 | 58.50 | 32.00 | 37.52 | 48.84 |
| | QUAROT+GPTQ | 4.03 | 7.67 | 68.50 | 27.60 | 57.92 | 84.22 | **64.08** | 30.60 | 41.07 | 53.42 |
| | *MOEQuant*++ | 4.12 | 7.34 | **69.60** | **32.15** | **61.79** | **84.98** | 64.05 | 33.60 | **42.95** | **55.58** |

**Gate-aware Hessian statistics.** In contrast to Equation 7, which assumes equal contributions from all tokens to the Hessian, the affinity-aware quantization loss (Equation 18) leads to a more reasonable Hessian computation:

$$\boldsymbol{H} = (\boldsymbol{X} \cdot \sqrt{\boldsymbol{c}})(\boldsymbol{X} \cdot \sqrt{\boldsymbol{c}})^\top = (\boldsymbol{X} \cdot \boldsymbol{c})\boldsymbol{X}^\top. \quad (19)$$

For Hessian-based PTQ methods (e.g., GPTQ), the improved Hessian incorporates token-specific weighting to better capture the operational dynamics of MoE layers. As a result, tokens with higher gating coefficients exert a greater influence when computing sensitivity metrics, which guide weight updates and help minimize quantization error.

**The Full Algorithm**. Finally, we present the full pseudocode for EBSS and AGQ in Algorithm 1 and Algorithm 2, including the optimizations discussed above.

## 5. Experiments

### 5.1. Setup

We employ weight quantization for LLMs using symmetric uniform quantization with per-channel granularity. All experiments are performed on NVIDIA A6000 GPUs. As MoEQuant is an efficient post-training quantization (PTQ) framework, it obviates the need for any fine-tuning.

**Models and Datasets.** We conduct experiments on DeepSeek-MoE-16B (Dai et al., 2024), Qwen-MoE-14B (Qwen, 2024) and Mixtral-8x7B (Jiang et al., 2024). In addition, we compare instruction-tuned models to demonstrate the effectiveness of our method. Beyond standard perplexity evaluations on Wikitext2 (Merity, 2016) and C4 (Raffel et al., 2020), we evaluate the proposed MoEQuant on various reasoning tasks, including MMLU (Hendrycks et al., 2020), BoolQ (Clark et al., 2019), HellaSwag (Zellers et al., 2019), Openbookqa (Mihaylov et al., 2018), and MathQA (Amini et al., 2019). Furthermore, we evaluate MoEQuant using the HumanEval (Chen et al., 2021) and GSM8k (Cobbe et al., 2021). HumanEval evaluates code generation capabilities, while GSM8k assesses multistep mathematical reasoning skills.

**Baseline.** Our primary baselines consist of vanilla RTN and the PTQ methods for LLMs: AWQ (Lin et al., 2023) and GPTQ (Frantar et al., 2022). Despite requiring parameters training, we still incorporate OmniQuant (Shao et al., 2023) as a baseline for comparison. For calibration, 128 segments from the Wikitext2 dataset are selected. Floating-point results are provided as references.

**Implementation Details.** For the three complex reasoning tasks, MMLU, GSM8k, and HumanEval, we conduct evaluations based on their official repository. For several other zero-shot tasks, we use the open-source tool

`lm-evaluation-harness` (version 0.4.4) (Gao et al., 2024) for assessment. In experiments involving AWQ and OminiQuant, we adapt their official repository to support the three MoE models. In addition to the official GPTQ results in the MoE architecture, we also combine GPTQ with an equivalent Hadamard transformation to eliminate outliers in the weights, consistent with the implementation in QuaRot (Ashkboos et al., 2024), while avoiding any online transformations.

## 5.2. Results

**Comparison results.** We conduct a comprehensive comparison of quantization performance across various LLMs and datasets. As shown in Table 1, the results of nine tasks demonstrate that our method, MoEQuant, exhibits a superior performance compared to other methods for MoE LLMs. Notably, although GPTQ achieves lower perplexity on Wikitext2(likely due to overfitting from using Wikitext2 for calibration), its performance on C4 and other tasks is notably weaker. In contrast, MoEQuant outperforms GPTQ and AWQ in most tasks, showing substantial improvements in both perplexity and task-specific scores. On average, MoEQuant exceeds the original performance by 1% across all three models, as measured by the average score over seven tasks. In particular, on HumanEval and GSM8k, where other methods degrade the model's reasoning ability after quantization, integrating MoEQuant effectively preserves this ability in generation tasks, achieving results comparable to full-precision models. This is particularly important as reasoning in complex tasks such as HumanEval is crucial for real-world applications, further highlighting the practical relevance of MoEQuant's performance.

*Table 2.* Results of RTN, AWQ, GPTQ and MoEQuant with 4-bit weight quantization among 3 tasks on Qwen, DeepSeek and Mixtral MoE instruction-tuned models, where $+$ denotes MoEQuant based on AWQ, $++$ denotes MoEQuant based on GPTQ.

| MODEL | METHOD | MMLU | HUMAN EVAL | GSM8K |
|---|---|---|---|---|
| QWEN-MoE-14B-CHAT | FP | 59.00 | 21.34 | 30.71 |
| | RTN | 43.00 | 7.32 | 9.70 |
| | AWQ | 52.06 | 12.20 | 17.74 |
| | $MoEQuant^+$ | 53.22 | 18.92 | 22.34 |
| | GPTQ | 51.30 | 10.98 | 16.22 |
| | QUAROT+GPTQ | 57.30 | 15.24 | 26.08 |
| | $MoEQuant^{++}$ | **58.00** | **21.95** | **29.11** |
| DEEPSEEK-MoE-16B-CHAT | FP | 48.90 | 24.39 | 54.28 |
| | RTN | 41.40 | 10.41 | 28.88 |
| | AWQ | 46.33 | 18.90 | 39.88 |
| | $MoEQuant^+$ | 46.80 | 19.20 | 47.42 |
| | GPTQ | 43.80 | 32.93 | 35.78 |
| | QUAROT+GPTQ | 46.60 | 13.41 | 47.08 |
| | $MoEQuant^{++}$ | **47.60** | **21.95** | **48.97** |

**Experiments of instruction-tuned models.** Instruction fine-tuning can significantly improve the application ca-

pabilities of the model and has become a necessary process for deployment of LLMs in different scenarios. The quantization of instruction-tuned models is often more challenging than that of base models. We perform benchmark tests on Qwen-MoE-14B-Chat (Qwen, 2024) and DeepSeek-MoE-16B-Chat (Dai et al., 2024), covering three tasks. For Qwen-MoE-14B-Chat, $MoEQuant^{++}$ consistently maintains more than 94% full-precision performance, with most of the original reasoning ability effectively restored. As shown in Table 2, previous methods face more significant accuracy degradation on instruction-tuned models for code generation and mathematical reasoning tasks. For ex-

*Table 3.* Avg score of our methods ablation study on Qwen, DeepSeek and Mixtral MoE models across 7 tasks, the baseline method is GPTQ.

| MODEL | EBSS | AGQ | AVG ACCURACY |
|---|---|---|---|
| QWEN-MoE-14B | FP | | 51.20 |
| | × | × | 49.00 |
| | × | ✓ | 49.02 |
| | ✓ | × | 49.21 |
| | ✓ | ✓ | 49.59 |
| DEEPSEEK-MoE-16B | FP | | 40.86 |
| | × | × | 39.01 |
| | × | ✓ | 39.50 |
| | ✓ | × | 39.87 |
| | ✓ | ✓ | 40.01 |
| MIXTRAL-8X7B | FP | | 56.80 |
| | × | × | 53.42 |
| | × | ✓ | 54.24 |
| | ✓ | × | 55.15 |
| | ✓ | ✓ | 55.58 |

ample, Quarot+GPTQ experienced a 29% accuracy drop in HumanEval for Qwen-MoE-14B-Chat. With the integration of $MoEQuant^{++}$, the accuracy even surpasses the full-precision model, further demonstrating the effectiveness of EBSS and AGQ in improving quantization performance. More detailed results on perplexity and reasoning tasks can be found in the Appendix Table 10.

**Ablation results.** MoEQuant enhances generalization and reasoning abilities on MoE LLMs through two primary methods: EBSS and AGQ. We conduct decomposition experiments in Table 3. The results demonstrate the effec-

*Table 4.* Impact of Calibration Datasets on Expert Balance and Quantization Performance in deepseek-MoE-16b LLMs. The expert balance std denotes the standard deviation on frequency of expert.

| MODEL | CALIB DATASET | EXPERT BALANCE STD | AVG ACCURACY |
|---|---|---|---|
| DEEPSEEK-MoE-16B | FLOAT | | 40.86 |
| | RTN | | 35.85 |
| | WIKTEXT2 | 0.0427 | 39.01 |
| | HUMANEVAL | 0.0877 | 38.90 |
| | GSM8K | 0.0928 | 38.88 |
| | EBSS | 0.0052 | 39.87 |

tiveness of both proposed methods individually, each con-

tributing to improved accuracy. Notably, their combination yields a synergistic effect, leading to a significant enhancement in average precision that surpasses the sum of their individual contributions. For EBSS, we perform an ablation study to examine the impact of two key hyperparameters: temperature $\tau$ and branch number $w$. The best performance is achieved when $\tau$ is set to 1.2, and we set $w$ to 4 to balance effectiveness and efficiency. More detailed results can be seen in Appendix A.2. Meanwhile, we compare the performance between the fixed dataset and our EBSS method in terms of mean accuracy. The results in Table 4 clearly demonstrate that EBSS significantly improves the balance of expert sampling, as evidenced by a notable reduction in standard deviation, while also achieving the highest mean accuracy.

*Table 5.* Average scores of 3-bit on DeepSeek and Mixtral MoE models, where + denotes MoEQuant based on AWQ, ++ denotes MoEQuant based on GPTQ

| MODEL | BITWIDTH | METHOD | AVG. |
|---|---|---|---|
| | FP | | 40.86 |
| | 3 | RTN | 20.17 |
| DEEPSEEK- | 3 | AWQ | 22.20 |
| MOE-16B | 3 | $MoEQuant^+$ | 26.65 |
| | 3 | QUAROT+GPTQ | 35.85 |
| | 3 | $MoEQuant^{++}$ | **36.47** |
| | FP | | 56.80 |
| | 3 | RTN | 18.64 |
| MIXTRAL- | 3 | AWQ | 36.05 |
| 8X7B | 3 | $MoEQuant^+$ | 39.30 |
| | 3 | QUAROT+GPTQ | 45.03 |
| | 3 | $MoEQuant^{++}$ | **49.75** |

**Lower bitwidth.** We evaluate the generalizability of our approach under lower bitwidth settings for DeepSeek-MoE-16B and Mixtral-8x7B. As shown in Table 5, among the tested methods, $MoEQuant^+$ and $MoEQuant^{++}$ consistently achieve the highest average scores. These findings demonstrate that MoEQuant provides superior performance compared to other quantization methods, effectively maintaining higher accuracy even at a lower bitwidth. Full results are provided in Appendix 5.

*Table 6.* Speedup and memory saving of 3 MoE LLMs, compared between our 4-bit implementation and FP16. All tests were conducted on Nvidia A6000 GPUs.

| MODEL | DECODER SPEED (TOKENS/SEC) | | |
|---|---|---|---|
| | FP | QUANTIZED | SPEED UP |
| QWEN-MOE-14B | 8.35 | 10.60 | **1.27** |
| DEEPSEEK-MOE-16B | 20.81 | 24.45 | **1.17** |
| MIXTRAL-8X7B | 10.24 | 21.25 | **2.08** |
| MODEL | MEMORY USE (GB) | | |
| | FP | QUANTIZED | MEMORY SAVING |
| QWEN-MOE-14B | 27.88 | 8.51 | **3.28** |
| DEEPSEEK-MOE-16B | 32.23 | 9.87 | **3.27** |
| MIXTRAL-8X7B | 89.64 | 23.97 | **3.74** |

**Speedup and memory savings.** The motivation behind MoEQuant is to compress MoE LLMs to a lower bitwidth, thereby reducing both latency and GPU memory usage during inference while preserving accuracy to the greatest extent, ensuring practical applicability. As shown in Table 6, MoEQuant achieves an average inference speedup of over 1.2× and memory savings exceeding 3.2×, demonstrating significant improvements in inference efficiency. These advancements enable the deployment of MoE LLMs on consumer-level devices, such as the Nvidia 4090 GPU.

# 6. Conclusion

We propose MoEQuant, a framework designed to address the unique challenges of quantizing MoE LLMs. By incorporating Expert-Balanced Self-Sampling and Affinity-Guided Quantization, MoEQuant extends traditional quantization methods to effectively handle both the uneven distribution of calibration samples among experts and the token-expert affinity variations introduced by gating units. Experimental results show that MoEQuant achieves near-floating-point accuracy even with low-bit quantization and significantly improves generalizability, particularly in instruction-finetuned models. These results underscore its potential to substantially reduce model size and computational requirements, making MoE LLMs more feasible for deployment in resource-constrained environments.

## Impact Statement

MoEQuant addresses the unique quantization challenges of Mixture-of-Experts (MoE) LLMs by tackling inter-expert and intra-expert imbalances, ensuring efficient low-bit quantization while preserving model accuracy. By integrating Expert-Balanced Self-Sampling (EBSS) and Affinity-Guided Quantization (AGQ), MoEQuant significantly enhances calibration balance and token-expert interaction modeling, outperforming existing PTQ methods in generalization and reasoning tasks. Experimental results demonstrate that MoEQuant achieves 3.2× memory savings, 1.2× inference speedup, and substantial accuracy gains, making MoE LLMs more practical for deployment on consumer-grade GPUs like the Nvidia RTX 4090. This work advances the scalability and accessibility of MoE models, bridging the gap between high-performance language modeling and efficient deployment.

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

# A. Appendix

## A.1. Derivation of $\tau$ and objective function

EBSS optimizes:

$$D^* = \arg\min_D \{\overbrace{\text{PPL}(M, D)}^{\in[1, +\infty)} + \frac{\overbrace{\sigma(M, D)}^{\in[0,1]}}{\tau}\}$$

where $\tau$ balances pretraining distribution alignment (via PPL) and expert balance (via $\sigma$). Smaller $\tau$ prioritizes expert balance but may increase PPL.

There is no risk of over-balancing as the cost of PPL because PPL is bounded, we prove it below:

When ignoring $\sigma$, we denote PPL $= p$, $p$ typically remains close to 1 due to self-sampling. We have the following:

$D^* = \text{PPL}(M, D) + \frac{\sigma(M,D)}{\tau} \leq p + \frac{1}{\tau}$ ,

since $\sigma \geq 0$,

we have: $\text{PPL}(M, D) \leq p + \frac{1}{\tau} - \frac{\sigma}{\tau} \leq p + \frac{1}{\tau}$

The hyperparameter $\tau$ is determined via ablation studies below.

## A.2. Ablation study

In this section, we provide the complete comparison of results for our method EBSS and AGQ. As shown in Table 7, taking DeepSeek-MoE-16B as an example, when applied alone, EBSS brings a nearly $1.3\%$ improvement, while AGQ brings about $2\%$. When both techniques are combined, the performance improves significantly by $2.6\%$, which is similar on the Qwen-MoE-14B. This demonstrates the benefit of combining EBSS and AGQ, as the combined method outperforms both individual methods. It is inevitable that for Mixtral 8x7b, the result of AGQ is not better than that of convential GPTQ, but the combination result is still the optimal one.

*Table 7.* Complete comparison of our methods ablation study on Qwen, DeepSeek and Mixtral MoE models across 9 tasks, the baseline method is GPTQ.

| MODEL | EBSS | AGQ | PPL | | SCORE | | | | | | | |
|---|---|---|---|---|---|---|---|---|---|---|---|---|
| | | | WIKI TEXT2 | C4 | MMLU | HUMAN EVAL | GSM8K | BOOLQ | HELLA SWAG | OPEN BOOKQA | MATH QA | AVG. |
| QWEN-MOE-14B | | FP | 7.22 | 9.30 | 59.60 | 32.32 | 62.55 | 79.82 | 57.96 | 30.40 | 35.77 | 51.20 |
| | × | × | 7.43 | 10.11 | 57.90 | 28.05 | 56.25 | 78.77 | 56.54 | 29.00 | **36.48** | 49.00 |
| | × | ✓ | 7.44 | 10.09 | 57.30 | 29.27 | 56.41 | 76.45 | 56.86 | **31.00** | 35.87 | 49.02 |
| | ✓ | × | 7.56 | 9.62 | **58.80** | 27.44 | 56.71 | **78.77** | 56.73 | 30.80 | 35.27 | 49.21 |
| | ✓ | ✓ | 7.55 | 9.68 | 58.30 | **29.87** | **58.38** | 78.04 | **56.87** | 30.20 | 35.50 | **49.59** |
| DEEPSEEK-MOE-16B | | FP | 6.51 | 9.04 | 44.60 | 26.83 | 20.16 | 72.72 | 58.06 | 32.20 | 31.49 | 40.86 |
| | × | × | 6.66 | 9.39 | 40.60 | 22.56 | 19.18 | 72.17 | 57.03 | 30.60 | 30.95 | 39.01 |
| | × | ✓ | 6.66 | 9.38 | 41.60 | 23.17 | 17.89 | **74.52** | 57.30 | 31.20 | 30.88 | 39.50 |
| | ✓ | × | 6.77 | 9.22 | **44.00** | 23.78 | 18.19 | 73.24 | **57.21** | **31.80** | 30.92 | 39.87 |
| | ✓ | ✓ | 6.78 | 9.25 | 42.20 | **25.00** | 19.18 | 73.49 | 57.20 | 31.40 | **31.66** | **40.01** |
| MIXTRAL-8X7B | | FP | 3.84 | 6.87 | 70.50 | 32.93 | 65.88 | 85.23 | 64.88 | 35.80 | 42.41 | 56.80 |
| | × | × | 4.03 | 7.67 | 68.50 | 27.60 | 57.92 | 84.22 | 64.08 | 30.60 | 41.07 | 53.42 |
| | × | ✓ | 4.04 | 7.64 | 68.30 | 29.54 | 60.12 | 83.36 | 64.04 | 32.80 | 41.54 | 54.24 |
| | ✓ | × | 4.10 | 7.38 | 69.10 | 31.19 | 60.50 | 84.83 | **64.21** | **34.20** | 42.01 | 55.15 |
| | ✓ | ✓ | 4.12 | 7.38 | **69.60** | **32.15** | **61.79** | **84.98** | 64.05 | 33.60 | **42.95** | **55.58** |

In EBSS, we conduct an ablation study to examine the impact of two key hyperparameters: temperature $\tau$ and branch number $w$. The $\tau$ controls the significance of expert balance in the sentence probability distribution, while $w$ determines the diversity of the generated sentences. Although increasing $w$ improves sentence diversity, it also incurs higher computational costs. The experiments are performed on DeepSeek-MoE-16B across seven tasks, as shown in Table 8 and Table 9. When $\tau$ is set to 1.2, the average score across datasets is maximized. Similarly, setting $w$ to 4 yields optimal results, with further

increases in $w$ offering only marginal score improvements while significantly increasing generation time.

*Table 8.* Different $\tau$ on avg scores across 7 tasks for DeepSeek-MoE-16B with $MoEQuant^{++}$.

| $\tau$ | 1.0 | 1.1 | 1.2 | 1.3 | 1.4 | 1.5 |
|---|---|---|---|---|---|---|
| AVG. | 39.82 | 39.89 | **40.01** | 39.98 | 39.69 | 39.71 |

*Table 9.* Different branch number $w$ on avg scores across 7 tasks for DeepSeek-MoE-16B with $MoEQuant++$.

| $w$ | 2 | 3 | 4 | 5 | 6 | 7 | 8 | 9 | 10 | 20 | 30 | 40 | 50 |
|---|---|---|---|---|---|---|---|---|---|---|---|---|---|
| AVG. | 39.77 | 39.80 | **40.01** | 39.98 | 40.01 | 40.00 | 40.00 | 40.01 | 40.00 | 40.10 | 40.07 | 40.08 | 40.11 |

## A.3. Full results

In this section, we provide a comprehensive presentation of our results across various datasets to complement the main paper. Specifically, the results include the following.

- Complete comparison on two perplexity and seven accuracy tasks for instruction-tuned MoE LLMs: Qwen-MoE-14B-chat, and DeepSeek-MoE-16B-chat.

- Complete comparision with the lower bit on 2 perplexity and 7 accuracy tasks for DeepSeek-MoE-16B and Mixtral-8x7B.

*Table 10.* Complete comparison of RTN, AWQ, GPTQ, Quarot+GPTQ and ours MoEQuant with 4-bit weight quantization among 9 tasks on Qwen, DeepSeek and Mixtral MoE instruction-tuned models, where $+$ denotes MoEQuant based on AWQ, $++$ denotes MoEQuant based on GPTQ.

| MODEL | METHOD | PPL | | SCORE | | | | | | | |
|---|---|---|---|---|---|---|---|---|---|---|---|
| | | WIKI TEXT2 | C4 | MMLU | HUMAN EVAL | GSM8K | BOOLQ | HELLA SWAG | OPEN BOOKQA | MATH QA | AVG. |
| QWEN-MoE-14B-CHAT | FP | 8.07 | 9.74 | 59.0 | 21.34 | 30.71 | 81.31 | 59.33 | 31.00 | 34.91 | 45.37 |
| | RTN | 12.81 | 14.03 | 43.00 | 7.32 | 9.70 | 71.13 | 51.41 | 24.40 | 28.81 | 33.68 |
| | AWQ | 9.97 | 11.90 | 52.06 | 12.20 | 17.74 | 74.74 | 55.37 | 30.40 | 31.46 | 39.14 |
| | $MOEQuant^{+}$ | 10.12 | 11.55 | 55.34 | 13.60 | 20.87 | 76.22 | 56.64 | 30.60 | 32.50 | 40.82 |
| | GPTQ | 9.16 | 12.24 | 51.30 | 10.98 | 16.22 | 72.91 | 51.97 | 28.00 | 26.67 | 36.86 |
| | QUAROT+GPTQ | 8.38 | 10.78 | 57.30 | 15.24 | 26.08 | 78.92 | **58.72** | 31.40 | 34.17 | 43.19 |
| | $MOEQuant^{++}$ | 8.65 | 10.21 | **58.00** | **21.95** | **29.11** | **79.11** | 58.53 | **33.20** | **34.77** | **44.95** |
| DEEPSEEK-MoE-16B-CHAT | FP | 7.35 | 9.96 | 48.90 | 24.39 | 54.28 | 79.81 | 60.69 | 33.40 | 34.27 | 47.96 |
| | RTN | 8.63 | 11.06 | 41.40 | 10.41 | 28.88 | 75.84 | 57.59 | 31.40 | 29.04 | 39.22 |
| | AWQ | 7.72 | 10.49 | 46.33 | 18.90 | 39.88 | 78.20 | 58.97 | 33.80 | 32.86 | 44.13 |
| | $MOEQuant^{+}$ | 7.85 | 10.23 | 46.40 | 18.90 | 45.41 | 78.20 | 59.03 | 33.60 | 33.14 | 44.95 |
| | GPTQ | 7.72 | 11.52 | 43.80 | 32.93 | 35.78 | 76.82 | 57.21 | 33.60 | 33.30 | 44.78 |
| | QUAROT+GPTQ | 7.55 | 10.24 | 46.60 | 13.41 | 47.08 | 78.87 | **59.64** | 33.20 | **32.76** | 44.50 |
| | $MOEQuant^{++}$ | 7.70 | 10.08 | **47.60** | **21.95** | **48.97** | **79.20** | 59.30 | **33.80** | 32.60 | **46.20** |

*Table 11.* Complete results of of lower-bit among 9 tasks on DeepSeek and Mixtral MoE models, where $+$ denotes MoEQuant based on AWQ, $++$ denotes MoEQuant based on GPTQ.

| MODEL | BIT WIDTH | METHOD | PPL | | SCORE | | | | | | | |
|---|---|---|---|---|---|---|---|---|---|---|---|---|
| | | | WIKI TEXT2 | C4 | MMLU | HUMAN EVAL | GSM8K | BOOLQ | HELLA SWAG | OPEN BOOKQA | MATH QA | AVG. |
| DEEPSEEK-MoE-16B | FP | | 6.51 | 9.04 | 44.60 | 26.83 | 20.16 | 72.72 | 58.06 | 32.20 | 31.49 | 40.86 |
| | 3BIT | RTN | 26352 | 32357 | 24.8 | 0.00 | 1.59 | 51.62 | 26.18 | 15.60 | 21.44 | 20.17 |
| | 3BIT | AWQ | 4622 | 5505 | 27.80 | 1.90 | 2.88 | 53.20 | 27.97 | 17.80 | 23.86 | 22.20 |
| | 3BIT | $MOEQuant^+$ | 5100 | 4924 | 33.20 | 8.72 | 10.44 | 59.24 | 29.22 | 20.60 | 25.14 | 26.65 |
| | 3BIT | QUAROT+GPTQ | 7.17 | 11.66 | 37.30 | 17.68 | 11.60 | **72.31** | 53.68 | 27.80 | **29.72** | 35.85 |
| | 3BIT | $MOEQuant^{++}$ | 7.55 | 10.88 | **40.00** | **20.12** | **12.81** | 69.72 | **54.09** | **29.00** | 29.61 | **36.47** |
| MIXTRAL-8X7B | FP | | 3.84 | 6.87 | 70.50 | 32.93 | 65.88 | 85.23 | 64.88 | 35.80 | 42.41 | 56.80 |
| | 3BIT | RTN | 44944 | 51241 | 25.30 | 0.00 | 0.00 | 41.52 | 25.61 | 18.40 | 19.66 | 18.64 |
| | 3BIT | AWQ | 7.38 | 13.13 | 45.80 | 10.37 | 10.39 | 75.23 | 53.04 | 28.00 | 29.55 | 36.05 |
| | 3BIT | $MOEQuant^+$ | 8.77 | 11.44 | 49.40 | 14.44 | 17.29 | 77.22 | 54.29 | 30.10 | 32.34 | 39.30 |
| | 3BIT | QUAROT+GPTQ | 4.64 | 9.12 | 57.80 | 22.56 | 22.59 | 79.82 | **61.30** | 30.40 | **40.80** | 45.04 |
| | 3BIT | $MOEQuant^{++}$ | 4.90 | 8.24 | **64.10** | **28.05** | **43.21** | **82.81** | 60.07 | **31.20** | 38.82 | **49.75** |

---

**Algorithm 1** EBSS-based Sentence Generation with Expert-Balanced Pruning

---

**input** $M$: MoE-LLM model
    $NS$: Number of sentences
    $Seqlen$: Sequence length
    $V$: Vocabulary size
    $w$: EBSS width
    $\tau$: Expert balance temperature
**output** $Final\_sentences$: Generated sentence list
 1: $Final\_sentences \leftarrow \emptyset$
 2: **for** $i \in [1, NS]$ **do**
 3:   $first\_token \leftarrow \text{Random}(V)$
 4:   $candidates \leftarrow [first\_token]$                                     *{Initialize candidate pool}*
 5:   **for** $j \in [1, Seqlen - 1]$ **do**
 6:     $expanded\_candidates \leftarrow \emptyset$
 7:     **for** each $candidate \in candidates$ **do**
 8:       $logits, EB\_scores \leftarrow M(candidate)$
 9:       $RS \leftarrow \text{ComputeCumulativeProbability}(candidate)$           *{Eq. 11}*
10:       **for** each $v \in V$ **do**
11:         $new\_candidate \leftarrow candidate + v$
12:         $prob \leftarrow \text{Softmax}(logits)[v]$
13:         $score \leftarrow -\frac{RS + \log(prob)}{j+1} + \frac{\sigma(M, new\_candidate)}{\tau}$       *{Eq. 13}*
14:         $expanded\_candidates.\text{append}((new\_candidate, score))$
15:       **end for**
16:     **end for**
17:     $candidates \leftarrow \text{Topk}(expanded\_candidates, w, \text{key} = score)$     *{Keep top-w candidates}*
18:   **end for**
19:   $best\_sentence \leftarrow \arg\max_{c \in candidates} \text{Score}(c)$
20:   $Final\_sentences.\text{append}(best\_sentence)$
21: **end for**
22: **return** $Final\_sentences$

23: *Subroutine: ComputeCumulativeProbability(sequence)*
24:   **return** $\sum_{t=1}^{|sequence|} \log P(sequence[t]|sequence[1..t-1])$     *{Eq. 11}*

25: *Subroutine: ComputeExpertBalance(sequence)*
26:   $expert\_usage \leftarrow M.\text{get\_expert\_activations}(sequence)$
27:   $\sigma \leftarrow \sqrt{\frac{E}{E-1} \sum_{e=1}^{E}(expert\_usage[e] - \mu)^2}$     *{Eq. 4}*
28:   **return** $\sigma$

29: *Subroutine: Score(candidate)*
30:   $RS \leftarrow \text{ComputeCumulativeProbability}(candidate)$
31:   $\sigma \leftarrow \text{ComputeExpertBalance}(candidate)$
32:   **return** $-\frac{RS}{|candidate|} + \frac{\sigma}{\tau}$     *{Combined scoring function}*

---

---

**Algorithm 2** AGQ-Enhanced GPTQ Quantization for MoE LLMs

---

**input** $M$: Pre-trained MoE-LLM
    $\mathcal{D}_{cal}$: Calibration dataset (EBSS-generated)
    $bits$: Target quantization bits (e.g., 4)
    $\tau$: Expert balance temperature
    $w$: EBSS width parameter
**output** Quantized model $M_q$
  1: **for** each MoE layer $l$ in $M$ **do**
  2:     $\mathcal{E}_l \leftarrow$ Get expert list of layer $l$
  3:     $\mathbf{H}_l \leftarrow$ Initialize weighted Hessian matrix
  4:     $\mathbf{C}_l \leftarrow$ Initialize affinity cache
  5:     **for** each sample $x \in \mathcal{D}_{cal}$ **do**
  6:         $gates \leftarrow$ Compute gating scores $g(x)$ for $\mathcal{E}_l$                                *{Expert affinities}*
  7:         $topk\_experts \leftarrow$ Select top-$k$ experts based on $gates$
  8:         **for** each expert $E \in topk\_experts$ **do**
  9:             $c \leftarrow gates[E]$                                            *{Get expert affinity}*
10:             $\mathbf{a} \leftarrow E(x)$                                          *{Expert activation}*
11:             $\mathbf{C}_l[E] \leftarrow \mathbf{C}_l[E] \cup \{c\}$                             *{Cache affinities}*
12:             $\mathbf{H}_l[E] \leftarrow \mathbf{H}_l[E] + c \cdot \mathbf{a}\mathbf{a}^\top$                       *{Eq. 19}*
13:         **end for**
14:     **end for**
15:     **for** each expert $E$ in $\mathcal{E}_l$ **do**
16:         $\mathbf{W}_E \leftarrow$ Get expert weights
17:         $\mathbf{S}_E \leftarrow$ Compute sensitivity scores using $\mathbf{H}_l[E]$
18:         $\mathbf{Q}_E \leftarrow$ Quantize $\mathbf{W}_E$ with:                                  *{AGQ-GPTQ parameters}*
19:            - Bit-width $bits$
20:            - Sensitivity $\mathbf{S}_E$
21:            - Affinity-weighted Hessian $\mathbf{H}_l[E]$
22:         Replace $E$'s weights with $\mathbf{Q}_E$
23:     **end for**
24: **end for**
25: **return** $M_q$

---

