# OpenReview forum: "MoEQuant: Enhancing Quantization for Mixture-of-Experts Large Language Models via Expert-Balanced Sampling and Affinity Guidance"
_ICML.cc/2025/Conference — ICML 2025 poster_

### Official Review · Reviewer_ZiDj · 2025-03-11

**Overall Recommendation:** 3

**Summary:**

This paper proposes a customized quantization method for MoE-LLMs, targeting two key challenges: inter-expert and intra-expert imbalance. To address these issues, the authors design a self-sampling approach to construct a balanced calibrated dataset and introduce an affinity-guided quantization loss to ensure balanced weight updates. Extensive experiments validate the effectiveness of the proposed method.

**Claims And Evidence:**

The claims in Lines 60-63 are not well supported by clear evidence. As shown in Table 1, previous method like GPTQ does not exhibit consistently poor performance across different datasets. The proposed method shows marginal improvements against GPTQ on several datasets.

**Essential References Not Discussed:**

NA

**Experimental Designs Or Analyses:**

The experiments are extensive and sound. However, several experiments are not well analyzed.
1. Even though the proposed method employs 3- or 4-bit weight quantization, the speedup observed in Table 4 is not significant. This questions the practical benefits of the proposed method. I'm not sure if it's due to the lack of activation quantization or expert routing overhead overshadow the benefits of weight quantization. Additionally, when combining MoEQuant with GPTQ, the paper does not report the corresponding latency. A comparison under this setting would be valuable for understanding the practical benefits.
2. The proposed method can be viewed as data-free quantization, it would be beneficial to give more detailed analyses about the synthetic data like sentence coherence and data diversity? I think this can increase the soundness of the proposed method.

**Methods And Evaluation Criteria:**

The proposed method addresses an important issue—MoE-LLM quantization. However, I have concerns regarding its scalability to larger MoE models, such as DeepSeek-V2. The paper lacks experiments on such models, leaving it unclear whether expert imbalance becomes more pronounced in larger architectures and how this might affect the performance of other post-training methods.

**Other Comments Or Suggestions:**

Previous post-training methods are mostly one-shot, requiring domain-specific datasets for fine-tuning. The proposed method uses self-sampling to bypass the need for collecting fine-tuning datasets. This approach eliminates the dependency on finetuning datasets, the method also offers advantages in terms of data privacy.

**Other Strengths And Weaknesses:**

The paper is well illustrated and the method design is clearly stated.
The paper studies a timely problem, which is to reduce the memory consumption of large MoE models without incurring significant performance degradation. It offers an efficient post-training quantization (PTQ) framework, and obviates the need for any fine-tuning.


## UPDATE AFTER REBUTTAL
My concerns about performance improvements and scalability are mostly addressed.  Quantization on MoE models is a timely research topic. Thus, I remain positive about this paper.

**Questions For Authors:**

The paper only demonstrates the existence of imbalance among experts in LLMs; however, it lacks empirical or theoretical analysis on how this imbalance affects the quantization process and results. Are there any analysis on the relationship between expert imbalance and quantization performance?
The code respiratory is not availiable.

**Relation To Broader Scientific Literature:**

The self-sampling method primarily relies on LLMs to generate synthetic data, a concept that has been explored in previous quantization, works, like [1]. The key difference here is that this method specifically selects data that ensures a more balanced expert utilization.
[1] LLM-QAT: Data-Free Quantization Aware Training for Large Language Models

The Affinity-Guided Quantization achieves a more reasonable Hessian computation by incorporating affinity scores provided by the gating network. Leveraging approximate Hessian information has been explored in previous work, like GPTQ.

**Theoretical Claims:**

There are no formal proofs in the paper. I have checked the formulas. Formulas (18,19) are based on the assumption that the Hessian matrix H  is Positive Semi-Definite. This assumption comes from GPTQ.

---

> ### Author Rebuttal · Authors · 2025-03-31
>
> We sincerely appreciate your constructive feedback. Below, we provide point-by-point responses, with all revisions incorporated accordingly.
>
> ---
> >The proposed method shows marginal improvements against GPTQ
>
> >When applying both proposed modules to QWENMOE-14B, the performance does not show a clear improvement
>
> While Table 1 shows modest overall gains (1.2–4.1%), **MoEQuant delivers substantial improvements in instruction-tuned models and complex multistep reasoning tasks(sensitive to quantization and critical for real applications).** As shown in [Table R1](https://anonymous.4open.science/r/MoEQuant-DDFD/Tables_for_R4_ZiDj.md):
> - On code generation(HumanEval) and mathe reasoning(GSM8K), MoEQuant achieves 4.7–9.8% relative accuracy gains in base models.
> - In instruction-tuned models—more representative of deployment scenarios—it achieves 17.2–23.6% gains.
>
> Additionly, we're sorry for potentially misleading readers about GPTQ in manuscript. As Line 353 states, it actually uses the initialization, similar to the competitive QuaRot.
> >However, I have concerns regarding its scalability to larger MoE models
>
> Table 1 already demonstrates MoEQuant's effectiveness on Mixtral-8x7B (56B params). To further address your concern, we provide results for DeepSeekV2 in [Table R2](https://anonymous.4open.science/r/MoEQuant-DDFD/Tables_for_R4_ZiDj.md), where MoEQuant achieves a consistent 3.75%+ relative gain.
> >the speedup observed in Table 4 is not significant
>
> The modest speedup for Qwen-MoE-14B and DeepSeek-MoE-14B stems from lower bandwidth pressure with fewer activated parameters, a trend also observed in [OSTQuant]. For Qwen-MoE-14B and DeepSeek-MoE-16B, where 2.7B and 2.8B params are active per step, acceleration remains low. **In contrast, Mixtral-8×7B achieves a 2.1× speedup. Notably, after quantization, Qwen-MoE-14B requires only 8.51 GB of memory, enabling MoE-LLM deployment on edge devices (e.g., RTX 3060).**
>
> [OSTQuant]:Refining Large Language Model Quantization with Orthogonal and Scaling Transformations
> >when combining MoEQuant with GPTQ, the paper does not report the corresponding latency
>
> We appreciate the your suggestion and add a comparison of time overhead in [Table R3](https://anonymous.4open.science/r/MoEQuant-DDFD/Tables_for_R4_ZiDj.md). Since EBSS reduces search complexity from exponential to linear through Probability-Guided Pruning and AGQ remains lightweight with negligible overhead, **the additional time increase is minimal(only 11 minutes for DeepSeek).**
> >it would be beneficial to give more analyses about the synthetic data
>
> We appreciate your constructive feedback and provide synthetic cases in [Table R4](https://anonymous.4open.science/r/MoEQuant-DDFD/Tables_for_R4_ZiDj.md), which highlight two key characteristics:
> 1. **Coherence and Distributional Alignment:** Generated sentences maintain coherence and align with the original models' distribution, also evidenced by consistently low PPL in Figure 4.
> 2. **Diversity:** The data spans diverse domains, including reasoning and code generation. This diversity alleviates expert imbalance and drives significant performance gains (relative 5–24%) on complex reasoning tasks(Table 2).
>
> > Paper only demonstrates the existence of imbalance among experts, however, it lacks empirical or theoretical analysis on how this imbalance affects the quantization results
>
> Thanks for your insightful feedback. we provide both empirical and theoretical analysis below:
> -  As shown in [Table R5](https://anonymous.4open.science/r/MoEQuant-DDFD/Tables_for_R4_ZiDj.md), **expert balance is positively correlated with final accuracy.** MoEQuant reduces imbalance by 90% compared to others, achieving the best accuracy by encouraging even token distribution among experts.
> - In GPTQ, the Hessian matrix quality directly impacts quantization precision. We estimate it via the finite difference method:
> $$H_{ij} \approx \frac{\partial^2{L}}{\partial w_i \partial w_j} \approx \frac{L(w+\epsilon e_i+ \epsilon e_j)-L(w + \epsilon e_i)-L(w+\epsilon e_j)+L(w)}{\epsilon^2}$$
> $\epsilon$ is the tiny perturbation, $e_i$ is the i-th standard basis vector, $L$ is the loss. A fourth-order Taylor expansion yields:
> $$\hat{H_{ij}} \approx H_{ij} + \frac{\epsilon^2}{12}(\nabla_{iiii}L+6\nabla_{iijj}L+\nabla_{jjjj}L)+O(\epsilon^4)$$ Assume the estimated variance of the loss is $Var(L)=\frac{\sigma^2}{n}$ , $\sigma^2$ is the population variance, and $n$ is the sample size. Then the expectation of the Hessian estimation:
> $$E[\hat{H_{ij}}] = H_{ij}+O(\epsilon^2)+O(\frac{\sigma^2}{n\epsilon^2})$$
> Then, the bias of the Hessian estimation is:
> $$Bias(\hat{H_{ij}}) = E[\hat{H_{ij}}]-H_{ij} \approx O(\epsilon^2)+O(\frac{\sigma^2}{n\epsilon^2})$$
> **The formula's second term is inversely proportional to sample size n: larger n means smaller statistical deviation, and vice versa. Thus, maintaining expert balance ensures all experts are well-calibrated, leading to better performance.**

---

> > ### Comment · Reviewer_ZiDj · 2025-04-07
> >
> > The rebuttal addresses my concerns regarding the method's rationale and its practical usage in reducing memory. However, the unavailability of the code raises concerns about its reproducibility. Overall, I am inclined to maintain the positive score.

---

> > > ### Author Response · Authors · 2025-04-07
> > >
> > > Dear Reviewer ZiDj,
> > >
> > > We sincerely thank you for your thorough review and positive feedback on our manuscript. We also understand your concern regarding the unavailability of the code.
> > >
> > > **We are pleased to inform you that the code is now ready and has been uploaded to an anonymized repository: [https://anonymous.4open.science/r/MoEQuant-DDFD/code/README.md](https://anonymous.4open.science/r/MoEQuant-DDFD/code/README.md).** The link will be included in the revised manuscript.
> > >
> > > The code is fully reproducible, but if you have any questions or need further clarification, please feel free to contact us. We look forward to your feedback.
> > >
> > > Best regards,
> > > Authors of MoEQuant

---

### Official Review · Reviewer_dN5E · 2025-03-11

**Overall Recommendation:** 4

**Summary:**

This paper proposes the MoEQuant framework, which aims to address the key challenges in the quantization of large language models (LLMs) with mixture of experts (MoE): inter-expert imbalance and intra-expert imbalance. Through two innovative methods, expert balanced self-sampling (EBSS) and affinity-guided quantization (AGQ), EBSS balances the expert utilization by self-generating calibration sets; AGQ introduces the affinity between samples and experts into the calculation of quantization error to optimize weight updates. Experiments show that MoEQuant significantly improves model performance under 4-bit quantization and achieves 1.2 times acceleration and 3.2 times compression in inference speed and memory usage, respectively, verifying its efficiency and universality.

## update after rebuttal
The Authors have addressed all of my concerns.

**Claims And Evidence:**

The following claims have no clear and  convincing evidence because no experiment study, theoretical proof and reference are provided.
" two primary challenges: (1) Inter-expert imbalance, referring to the uneven distribution of samples across experts, which leads to insufficient and biased calibration for less frequently utilized experts; (2) Intra-expert imbalance, arising from MoE’s unique aggregation mechanism, which leads to varying degrees of correlation between different samples and their assigned experts."

**Essential References Not Discussed:**

GW-MoE (Global Workspace Theory for MoE Routers) is a recent work which introduces a method to mitigate router uncertainty by broadcasting tokens across experts during fine-tuning. While the paper addresses uncertainty implicitly, GW-MoE provides a direct and complementary approach that could enhance the robustness of the proposed methods.
At the same time, the article can introduce how MoE currently solves the load imbalance of inter-experts and intra-experts.

**Experimental Designs Or Analyses:**

yes

**Methods And Evaluation Criteria:**

yes

**Other Comments Or Suggestions:**

1. While EBSS aims to balance expert activation during calibration, it may inadvertently bias the model towards easier expert routes during self-sampling. Suggest adding a mechanism to explicitly control the trade-off between perplexity minimization and expert balance (e.g., via a dynamic weighting factor for the σ term in Equation 9).
2. The current AGQ formulation assumes linear propagation of affinities through experts (Equation 17). However, non-linear activation functions (e.g., ReLU) and residual connections in expert networks may break this assumption. Propose extending AGQ to account for non-linear interactions (e.g., via layer-wise affinity masks).

**Other Strengths And Weaknesses:**

Strengths：
1. The EBSS and AGQ methods proposed in this article effectively solve the shortcomings of the MoE model quantization compression that leads to a sharp loss of performance. At the same time, the above methods can also be used to solve the problem of unbalanced load between experts and within experts in the general MoE model, optimize expert training, and improve resource utilization efficiency.
2. As a plug-in framework, MoEQuant seamlessly integrates existing PTQ methods (such as AWQ and GPTQ), and can improve the quantitative effect of the MoE model without modifying the underlying code.

Weaknesses：
1. EBSS's self-sampling relies on probability-guided path pruning, which requires maintaining candidate sequence branches and calculating cumulative probabilities, which may introduce additional computational burden, especially limiting efficiency when generating long sequences.
2. The performance of the method is highly dependent on hyperparameters (such as temperature τ and number of branches w). Although the paper gives the optimal parameters (τ=1.2, w=4), in actual deployment, fine-tuning is required for different models, which increases the complexity of use.
3. AGQ assumes that the linear relationship between samples and experts is approximately valid (Equation 17), but in extremely sparse or nonlinear routing scenarios, affinity modeling may not be able to fully capture the complexity of interactions between experts, resulting in residual quantization errors.

**Questions For Authors:**

1. How does EBSS dynamically adjust the trade-off between minimizing perplexity and balancing expert activations? Specifically, the paper mentions using a temperature parameter τ (Equation 9) to control this balance, but the optimal τ=1.2 is fixed in experiments. How was τ=1.2 chosen, and is there a risk of over-balancing experts at the cost of perplexity? If τ is highly model-dependent or requires extensive tuning, this could limit EBSS’s practicality. Conversely, if τ can be generalized across models, it strengthens EBSS’s appeal.
2. While the paper fixes τ=1.2 and w=4, other hyperparameters (e.g., pruning threshold for candidate sequences) also will influence performance. Can you provide a sensitivity analysis of all EBSS hyperparameters (e.g., temperature range, branching factor) across different model scales (e.g., 14B vs. 6B parameters)? If EBSS requires case-by-case hyperparameter tuning, its adoption cost increases.

**Relation To Broader Scientific Literature:**

1. MoE Architecture and Core Components: The paper's discussion of MoE's modular structure, comprising router and experts, aligns with foundational works that systematically explain how MoE decomposes tasks and allocates them to specialized sub-networks. This modular design has been a cornerstone in scaling large models while maintaining efficiency.
2. Efficiency and Specialization: The paper highlights MoE's advantages in computational efficiency (Mixtral 8x22B's sparse activation) and specialized expert roles (handling specific token types like punctuation or visual data) . These ideas build upon prior research demonstrating MoE's superiority over dense models in resource utilization.
3. Challenges and Mitigation: The paper addresses challenges such as load imbalance and router uncertainty, which are critical issues in MoE training and inference. For instance,  proposes a method (GW-MoE) to resolve uncertainty in router decisions by leveraging global workspace theory, a concept not explicitly discussed in the paper but relevant to its context.

**Theoretical Claims:**

This paper does not provide direct experimental or theoretical proof that the poor MoE quantization effect is caused by load imbalance.

---

> ### Author Rebuttal · Authors · 2025-03-31
>
> We sincerely appreciate your constructive feedback. Below, we provide point-by-point responses, with all revisions incorporated accordingly.
>
> ---
> >paper does not provide direct experimental or theoretical proof that the poor MoE quantization effect is caused by load imbalance
>
> We provide additional theoretical and experimental justifications below:
> - Inter-Expert Imbalance
>   - Theoretical Basis: [Ours derivation](https://anonymous.4open.science/r/MoEQuant-DDFD/asserts/inter-expert_balance.md) proves that expert balance is helpful for accurate Hessian.
>   - Experimental: [Table R1](https://anonymous.4open.science/r/MoEQuant-DDFD/Tables_for_R3_dN5E.md) shows a positive correlation between expert balance and accuracy. EBSS(Expert-Balanced Self-Sampling) achieves 10x imbalance reduction and best accuracy. Table 5 shows EBSS reduces the accuracy gap by 49%.
> - Intra-Expert Imbalance
>   - Theoretical: Eq 15 formalizes expert aggregation, showing token-expert affinity varies. Eq 17 leverages linear equivalence for layer-wise quantization to anticipate affinity shifts.
>   - Experimental: Table 5 shows that on DeepSeek-MoE-16B, AGQ(Affinity-Guided Quantization) alone reduces the 'Quarot+GPTQ' vs. FP gap by 30%, and combined with EBSS, by 60%.
>
> >Supplementary Material
>
> We provide pseudocode for EBSS and AGQ in [Algorithm 1-2](https://anonymous.4open.science/r/MoEQuant-DDFD/Tables_for_R3_dN5E.md).
>
> Experiments cover code generation(HumanEval) and complex math reasoning(GSM8K). Table 2 shows we improve HumanEval over others by 44%-63%.
>
> Figure 2 shows that EBSS significantly reduces inter-expert imbalance.
> >GW-MoE is not explicitly discussed in the paper but relevant to its context
>
> We appreciate GW-MoE's insights in identifying routing uncertainty, which highlights load balancing. We've incorporated this into our revision for greater scholarly depth.
> >EBSS may introduce additional computational burden
>
> We appreciate your concern, but EBSS reduces complexity significantly through three innovations:
> - **Probability-Guided Path Pruning** eliminates unlikely branches using cumulative probability and expert balance, reducing candidates from $V^n$(exponential) to $mn$(linear).
> - **Cumulative Probability** stores only $m$(e.g., 4) scalars, cutting storage overhead from $mV$ to $m$, enabling efficient PPL computation via Eq 12.
> - **Deferred Imbalance Calculation** avoids iterating over all tokens (e.g., 150K) by leveraging known expert distributions from the current sequence.
>
> Experiments in [Table R6](https://anonymous.4open.science/r/MoEQuant-DDFD/Tables_for_R3_dN5E.md) shows EBSS's efficency(just 17–42 mins).
> >How does EBSS adjust the trade-off between PPL and expert balance? is there a risk of over-balancing experts at the cost of PPL?
>
> EBSS optimizes: $$D^* = \arg\min_{D}\{
> \overbrace{\text{PPL}(M,D)}^{\in[0,1]} + \overbrace{\frac{\sigma(M,D)}{\tau}}^{\in[1,+\infty)}
>  \}$$where $\tau$ balances data distribution alignment (via PPL) and expert balance (via $\sigma$). **Smaller τ prioritizes expert balance but may increase PPL.**
>     $\tau=1.2$ is determined through ablation in the manuscript. [Table R2](https://anonymous.4open.science/r/MoEQuant-DDFD/Tables_for_R3_dN5E.md) further validate its generalizability in more models.
>
> **There is no risk of over-balancing as the cost of PPL because PPL is bounded, we prove it below:**
> * when ignoring $\sigma$，we denote $\text{PPL}=p$, $p$ typically remains close to 1 due to self-sampling(see Fig 4).   We have:$$D^* = \text{PPL}({M,D})+ \frac{\sigma(M,D)}{\tau}\leq p+\frac{1}{\tau}$$ since $\sigma \geq 0$, we have: $$\text{PPL}(M,D)\leq p+\frac{1}{\tau} - \frac{\sigma}{\tau}\leq p+\frac{1}{\tau}$$
>
> >Can you provide a sensitivity analysis of all EBSS hyperparameters
>
> Additional ablations in [Table R2-R3](https://anonymous.4open.science/r/MoEQuant-DDFD/Tables_for_R3_dN5E.md), which shows：
> -  **τ = 1.2 is also generalizable and effective on other models.**
> - Accuracy improves with increasing $w$, though marginal gains diminish significantly beyond **w=4, offering an effective trade-off between performance and efficiency.**
>
> >non-linear activation functions and residual connections in expert networks may break this assumption
>
> For nonlinear SiLU in MoE LLMs, Eq 17 and AGQ remain effective. [Table R4 and R5](https://anonymous.4open.science/r/MoEQuant-DDFD/Tables_for_R3_dN5E.md) verify them from respectively:
> - **Numerical Error.** The reformulated gate output:$$y_i\approx\left((xW^{up})\odot f(c_ixW^{gate})\right)W^{down}$$ remains high cosine similarity(higher c close to 1) with true outputs. Moreover, top-k selection inherently prioritizes high-affinity tokens, further minimizing approximate error.
> - **Task Performance.** Applying AGQ to gate layers yields significant gains in both PPL and accuracy(1.2%+ gain), confirming its effectiveness.
>
> AGQ and residual connections (outside MoE layers) are decoupled. We agree that affinity masks hold promise and will explore it as MoE evolves.

---

> > ### Comment · Reviewer_dN5E · 2025-04-09
> >
> > I read the rebuttal, and I will raise my score accordingly.

---

> > > ### Author Response · Authors · 2025-04-09
> > >
> > > Dear Reviewer dN5E,
> > >
> > > Thank you for your constructive comments and valuable suggestions. We will include them in the final manuscript.
> > >
> > > We are glad to address all your questions and sincerely appreciate your recognition.
> > >
> > >
> > > Best regards,
> > >
> > > Authors of MoEQuant

---

### Official Review · Reviewer_NkUk · 2025-03-12

**Overall Recommendation:** 3

**Summary:**

MoEQuant is a novel quantization framework designed for Mixture-of-Experts (MoE) large language models (LLMs). It addresses the challenges of accuracy degradation encountered in traditional post-training quantization (PTQ) methods. The framework introduces two techniques: Expert-Balanced Self-Sampling (EBSS) and Affinity-Guided Quantization (AGQ). EBSS efficiently constructs balanced calibration datasets, while AGQ incorporates expert-sampling affinities into the quantization process. Experimental results demonstrate significant performance gains and efficiency improvements, making MoE models more practical for deployment on resource-constrained devices.

**Claims And Evidence:**

There is clear and convincing evidence of the claims made in the submission.

**Essential References Not Discussed:**

[1] Shao, Wenqi, et al. "Omniquant: Omnidirectionally calibrated quantization for large language models." arXiv preprint arXiv:2308.13137 (2023).

[2] Chen, Mengzhao, et al. "Efficientqat: Efficient quantization-aware training for large language models." arXiv preprint arXiv:2407.11062 (2024).

**Experimental Designs Or Analyses:**

The analysis of Quantization on MoE model is convinced, the datasets and benchmarks choosed in this paper are common.

**Methods And Evaluation Criteria:**

The proposed methods and evaluation criteria make sense for the deployment of Mixture-of-Experts large language models.

**Other Comments Or Suggestions:**

None

**Other Strengths And Weaknesses:**

Strengths:
1. MoEQuant introduces two innovative methods—Expert-Balanced Self-Sampling (EBSS) and Affinity-Guided Quantization (AGQ)—which address the unique challenges of quantizing MoE models, such as inter-expert and intra-expert imbalances.

2. The framework achieves substantial performance gains, particularly in complex reasoning tasks like HumanEval, and enhances efficiency by reducing model size and computational requirements. This makes MoE models more practical for real-world applications.

3. MoEQuant significantly improves the generalization ability of MoE models, especially when dealing with instruction-tuned models. It demonstrates robust performance across various tasks and datasets, outperforming conventional PTQ methods.

4. By effectively compressing MoE models to lower bitwidths, MoEQuant facilitates the deployment of these models on consumer-grade devices with limited memory and computational resources, thus broadening their accessibility and scalability.

Weaknesses：

1. The study focuses solely on weight quantization and does not address activation quantization. This limited scope may overlook potential benefits and challenges associated with activations, which could be critical for overall model efficiency and performance.

2. MoEQuant is primarily an enhancement of existing quantization methods like AWQ and GPTQ. While it introduces novel techniques, it does not represent a fundamentally new approach. This raises questions about the originality and potential for significant breakthroughs in quantization strategies.

3. The experimental results are not much competitive, and the baselines are not the latest, and AWQ+GPTQ would be much better. And the ablation study in Table 5 shows that EBSS and AGQ improve the accuracy of the quantized model very little

**Questions For Authors:**

1. Could you provide comparison of the time cost for MoEQuant and baselines?

2. Are the router layers of MoE quantized?  how do these layers affect the classification accuracy of MoE and the performance of the quantized model?

3. Could you shows the latest weight-only quantization methods on MoE models, such as OmniQuant [1] and EfficientQAT [2].

[1] Shao, Wenqi, et al. "Omniquant: Omnidirectionally calibrated quantization for large language models." arXiv preprint arXiv:2308.13137 (2023).

[2] Chen, Mengzhao, et al. "Efficientqat: Efficient quantization-aware training for large language models." arXiv preprint arXiv:2407.11062 (2024).

**Relation To Broader Scientific Literature:**

Previous LLM-PTQ methods (such as AWQ, GPTQ) focus on dense large language models, this paper proposes method for MoE-LLMs, which are different.

**Theoretical Claims:**

This paper do not contains any  theoretical claims. In the section of  Expert-Balanced Self-Sampling,  this paper presents the optimization process, which is enough to demonstarte the challenge.

---

> ### Author Rebuttal · Authors · 2025-03-31
>
> We sincerely appreciate your constructive feedback. Below, we provide point-by-point responses, with all revisions incorporated accordingly.
>
> ---
> >The study focuses solely on weight quantization and does not address activation quantization.
>
> Memory constraints are the decisive factor for deploying MoE LLMs, which require orders of magnitude more memory than dense models (e.g., DeepSeek-MoE-14B retains 14B params in memory despite activating only 2.7B). **Therefore, we prioritize weight quantization because it significantly alleviates memory pressure,** achieving a 3.74× memory saving and a 2× speedup (see Table 4) for Mixtral-8x7B, thereby enabling deployment on consumer GPUs.
>
> To further address your concern, we add W4A4 experiments. [Table R1](https://anonymous.4open.science/r/MoEQuant-DDFD/Tables_for_R2_NkUk.md) show that MoEQuant improves accuracy by over 3% compared to QuaRot, demonstrating its versatility across diverse quantization scenarios.
> >MoEQuant is primarily an enhancement of existing quantization methods like AWQ and GPTQ. While it introduces novel techniques
>
> MoEQuant is a novel PTQ framework tailored for MoE LLMs rather than an enhancement, addressing key limitations of existing methods through three contributions:
> - **Problem Identification:** As shown in Table 1, previous PTQ methods to MoE LLMs causes severe accuracy drops(up to 17%), We attribute it to 2 key issues: inadequate calibration across experts(inter-expert imbalance) and inaccurate quantization error modeling(intra-expert imbalance) inherent in MoE quantization.
> - **Technical Innovation:** EBSS(Expert-Balanced Self-Sampling) ensures adequate expert calibration via probability-guided pruning and deferred imbalance evaluation, maintaining alignment with fp models while reducing search complexity (Sec. 4.2). AGQ(Affinity-Guided Quantization) incorporates token-expert affinities into quantization loss and Hessian statistics, enabling precise sensitivity analysis for MoE’s dynamic routing (Sec. 4.3).
> - **Empirical Validation:** MoEQuant seamlessly integrates with existing PTQ methods(AWQ,GPTQ,QuaRot), significantly improving accuracy. As validated in [Table R7](https://anonymous.4open.science/r/MoEQuant-DDFD/Tables_for_R2_NkUk.md), it outperforms QuaRot by 5–23% on complex tasks like HumanEval and GSM8K, effectively closing prior performance gaps.
>
> >The baselines are not the latest and AWQ+GPTQ would be much better
>
> We would like to clarify that, as described in Line 383, our GPTQ implementation equal 'QuaROt+GPTQ', a current SOTA training-free PTQ method, this ensures rigorous baseline.
>
> To further address your concern, we have added a comparison with Omniquant in [Table R4-6](https://anonymous.4open.science/r/MoEQuant-DDFD/Tables_for_R2_NkUk.md) and MoEQuant still works better.  We also manually implemented GPTQ+AWQ but obtained worse performance.
> > the ablation study in Table 5 shows that EBSS and AGQ improve the accuracy very little
>
> While Table 5 shows modest overall gains(1.2–4.1% relative), we emphasize that **MoEQuant delivers significant gains in instruction-tuned models and complex reasoning tasks(sensitive to quantization and critical for real applications).** As shown in [Table R2](https://anonymous.4open.science/r/MoEQuant-DDFD/Tables_for_R2_NkUk.md):
> - On code generation(HumanEval) and mathematical reasoning(GSM8K) , MoEQuant achieves 4.69–9.84% relative accuracy gains in non-instruction-tuned models.
> - For instruction-tuned models, which better reflect deployment scenarios, MoEQuant demonstrates 17.22–23.57% relative gains.
>
> > Could you provide comparison of the time cost for MoEQuant and baselines?
>
> We add a comparison of the time costs for MoEQuant and GPTQ. As shown in [Table R3](https://anonymous.4open.science/r/MoEQuant-DDFD/Tables_for_R2_NkUk.md), Due to the incorporation of three cost-reducing techniques in EBSS—namely Probability-Guided Path Pruning, Cumulative Probability, and Deferred Expert Imbalance Calculation—**MoEQuant only increases a small amount of time overhead compared to GPTQ.**
> >Are the router layers of MoE quantized? how do these layers affect the performance of the quantized model?
>
> Router layers are quantized. We did not find an obvious impact on the final performance. This may be due to the robustness of gating and the redundancy of expert knowledge.
> >Could you show the latest weight-only quantization methods on MoE models, such as OmniQuant and EfficientQAT
>
> In [Table R4-R6](https://anonymous.4open.science/r/MoEQuant-DDFD/Tables_for_R2_NkUk.md), we provide additional results comparing MoEQuant with OmniQuant. **While a direct comparison may be unfair since MoEQuant is designed for PTQ and OmniQuant involves training, MoEQuant consistently outperforms OmniQuant across all tested models.** This advantage arises because OmniQuant relies on a fixed calibration set, disregarding expert balance. EfficientQAT, being a global QAT algorithm, is beyond the scope of our comparison.

---

> > ### Comment · Reviewer_NkUk · 2025-04-07
> >
> > The anonymous link doesn't seem to be accessible, could you check it?

---

> > > ### Author Response · Authors · 2025-04-07
> > >
> > > Dear Reviewer NkUk,
> > >
> > > We have made the table available via an anonymous [link](https://anonymous.4open.science/r/MoEQuant-DDFD/Tables_for_R2_NkUk.md), though accessibility may vary due to potential network issues. For convenience, we have also reorganized the key results below for direct reference.
> > >
> > > **Table R1: Comparison of W4A4 Quantization Performance of MoEQuant and Quarot on 3 MoE LLMs.**
> > >
> > > |Model|Method|WikiText2 PPL|Avg Accuracy|
> > > |-|-|:-:|:-:|
> > > |Qwen-MoE-14B|Float||51.22|
> > > ||QuaRot+GPTQ|8.40|46.30|
> > > ||MoEQuant|8.54|**48.62**|
> > > |DeepSeek-MoE-16B|Float|6.51|40.86|
> > > ||QuaRot+GPTQ|7.82|35.33|
> > > ||QuaRot+MoEQuant|7.90|**37.84**|
> > > |Mixtral-8x7B|Float|3.84|56.80|
> > > ||QuaRot+GPTQ|4.92|50.22|
> > > ||QuaRot+MoEQuant|5.03|**53.15**|
> > >
> > > **Table R2: Performance of Different Quantization Methods on MoE LLMs across two Multi-step Reasoning Tasks (HumanEval and GSM8k).** "Gain" means the improvement of the current method compared to the previous method.
> > >
> > > |MODEL|METHOD|HuamnEval|GSM8K|AVG Accuracy|Gain|
> > > |-|:-:|:-:|:-:|:-:|:-:|
> > > |QWEN-MoE-14b-CHAT|FP|21.34|30.71|26.03|-|
> > > ||RTN|7.32|9.70|8.51|-|
> > > ||GPTQ|10.98|16.22|13.60|-|
> > > ||Quarot+GPTQ|15.24|26.08|20.66|-|
> > > ||MoEQuant++|**21.95**|**29.11**|**25.53**|23.57%|
> > > |DEEPSEEK-MoE-16b-CHAT|FP|24.39|54.28|39.34|-|
> > > ||RTN|10.41|28.88|19.65|-|
> > > ||GPTQ|10.93|35.78|23.36|-|
> > > ||Quarot+GPTQ|13.41|47.08|30.25|-|
> > > ||MoEQuant++|**21.95**|**48.97**|**35.46**|17.22%|
> > > |QWEN-MoE-14b|FP|32.32|62.55|47.44|-|
> > > ||RTN|14.63|16.07|15.35|-|
> > > ||GPTQ|20.73|22.82|21.77|-|
> > > ||Quarot+GPTQ|28.05|56.25|42.15|-|
> > > ||MoEQuant++|**29.87**|**58.38**|**44.13**|4.69%|
> > > |DEEPSEEK-MoE-16b|FP|26.83|20.16|23.50|-|
> > > ||RTN|18.90|10.54|14.72|-|
> > > ||GPTQ|21.34|11.60|16.47|-|
> > > ||Quarot+GPTQ|22.56|19.18|20.87|-|
> > > ||MoEQuant++|**25.00**|**19.18**|**22.09**|5.85%|
> > > |MIXTRAL-8x7B|FP|32.93|65.88|49.41|-|
> > > ||RTN|28.05|27.90|27.98|-|
> > > ||GPTQ|24.39|42.15|24.27||
> > > ||Quarot+GPTQ|27.60|57.92|42.76|-|
> > > ||MoEQuant++|**32.15**|**61.79**|**46.97**|9.84%|
> > >
> > > **Table R3: Time Cost Comparison of GPTQ and MoEQuant.** All test are conducted on a single A800 GPU.
> > >
> > > |Model|Method|Time Cost|
> > > |-|-|-|
> > > |Qwen-MoE-14B|GPTQ|37 mins|
> > > ||MoEQuant|54 mins|
> > > |DeepSeek-MoE-16B|GPTQ|41 mins|
> > > ||MoEQuant|52 mins|
> > > |Mixtral-8x7B|GPTQ|73 mins|
> > > ||MoEQuant|115 mins|
> > >
> > > **Table R4: 4-bit Quantization Performance of OmniQuant and MoEQuant on Qwen-MoE-14B.**
> > >
> > > |Method|WikiText2 PPL ↓|C4 PPL ↓|MMLU|HumanEval|GSM8K|BoolQ|Hellaswag|OpenBookQA|MathQA|AVG Accuracy|
> > > |-|-|-|-|-|-|-|-|-|-|-|
> > > |FP|7.22|9.30|59.60|32.32|62.55|79.82|57.96|30.40|35.77|51.20|
> > > |OmniQuant|7.67|9.98|56.30|31.71|52.39|78.20|56.58|29.40|33.63|48.31|
> > > |MoEQuant|7.55|9.62|58.30|29.87|58.38|78.04|56.87|30.20|35.50|**49.59**|
> > >
> > >
> > > **Table R5: 4-bit Quantization Performance of OmniQuant and MoEQuant on DeepSeek-MoE-16B.**
> > >
> > > |Method|WikiText2 PPL ↓|C4 PPL ↓|MMLU|HumanEval|GSM8K|BoolQ|Hellaswag|OpenBookQA|MathQA|AVG Accuracy|
> > > |-|-|-|-|-|-|-|-|-|-|-|
> > > |FP|6.51|9.04|44.60|26.83|20.16|72.72|58.06|32.20|31.49|40.86|
> > > |OmniQuant|6.79|9.49|43.50|21.95|18.65|73.82|56.67|32.40|31.02|39.72|
> > > |MoEQuant|6.78|9.22|42.20|25.00|19.18|73.49|57.20|31.40|31.66|**40.01**|
> > >
> > > **Table R6: 4-bit Quantization Performance of OmniQuant and MoEQuant on Mixtral-8x7B.**
> > >
> > > |Method|WikiText2 PPL ↓|C4 PPL ↓|MMLU|HumanEval|GSM8K|BoolQ|Hellaswag|OpenBookQA|MathQA|AVG Accuracy|
> > > |-|-|-|-|-|-|-|-|-|-|-|
> > > |FP|3.84|6.87|70.50|32.93|65.88|85.23|64.88|35.80|42.41|56.80|
> > > |OmniQuant|4.19|7.20|68.10|34.75|57.01|84.13|63.03|33.00|41.91|54.56|
> > > |MoEQuant|4.12|7.34|69.60|32.15|61.79|84.98|64.05|33.60|42.95|**55.58**|
> > >
> > > **Table R7: Results of RTN, Omniquant, AWQ, GPTQ, Quarot+GPTQ and ours MoEQuant with 4-bit Quantization among 9 Tasks on DeepSeekMoE-16B and Mixtral-8x7B.** where + denotes MoEQuant based on AWQ, ++ denotes MoEQuant based on Quarot+GPTQ.
> > >
> > > |Model|Method|WikiText2 PPL↓|C4 PPL↓|MMLU|HumanEval| GSM8K|BoolQ|Hellaswag|OpenBookQA|MathQA|AVG Accuracy|
> > > |-|-|-|-|-|-|-|-|-|-|-|-|
> > > |DeepSeek-MoE-16B|FP|6.51|9.04| 44.60|26.83|20.16|72.72|58.06|32.20| 31.49|40.86|
> > > ||RTN| 7.47|10.01|36.10|18.90|10.54|70.21|55.76|30.60|28.87|35.85|
> > > ||OmniQuant|6.79|9.49| 43.50|21.95|18.65|73.82|56.67|32.40|31.02|39.72|
> > > ||AWQ|6.80|9.50|40.57|25.00|17.06|71.65|56.42|32.20|31.76|39.23|
> > > ||MoEQuant+| 6.94|9.32|41.20|25.00|18.90|71.98|56.79|32.12|31.82|39.68|
> > > ||GPTQ|6.82|10.35|39.60|21.34|11.60|72.14|56.05|30.60|30.35|37.38|
> > > ||Quarot+GPTQ|6.66|9.39|40.60|22.56|19.18|72.17|57.03|30.60|30.95|39.01|
> > > ||MoEQuant++|6.78|9.22|42.20|25.00|19.18|73.49|57.20|31.40|31.66|**40.01**|
> > > |Mixtral-8x7B|FP|3.84|6.87|70.50|32.93|65.88|85.23|64.88|35.80|42.41 |56.80|
> > > ||RTN|5.41|8.13|62.20|28.05|27.90|80.85|61.73|32.20|37.35|47.18|
> > > ||OmniQuant|4.19|7.20| 68.10|  34.75|57.01|84.13|63.03|33.00|41.91|54.56|
> > > ||AWQ|5.01|7.98|62.75|25.00|38.67|79.97|62.11|33.60|38.43|48.64|
> > > ||MoEQuant+| 5.15|7.84|64.66|25.45|50.66|81.03|62.73|34.00|39.77|51.19|
> > > ||GPTQ|4.84|8.08|64.30|24.39|42.15|83.03|58.50|32.00|37.52|48.84|
> > > ||Quarot+GPTQ|4.03|7.67|68.50|27.60|57.92|84.22|64.08|30.60|41.07|53.42|
> > > ||MoEQuant++|4.12|7.34|69.60|32.15|61.79| 84.98 |64.05|33.60|42.95 |**55.58**|

---

### Official Review · Reviewer_eTCn · 2025-03-15

**Overall Recommendation:** 3

**Summary:**

This paper introduces a framework named MoEQuant which designed to efficiently quantize MoE large language models while addressing calibration imbalances. It tackles two key challenges: the unequal distribution of calibration samples across different experts, and the varying affinities between tokens and their assigned experts. By integrating Expert-Balanced Self-Sampling to generate a balanced calibration dataset and Affinity-Guided Quantization to incorporate token-expert relationships into the quantization process, MoEQuant significantly reduces quantization errors. Extensive experiments demonstrate that MoEQuant not only maintains near-full-precision accuracy under low-bit quantization but also achieves faster inference speeds and substantial memory savings.

## Update after rebuttal.
The additional explanation has indeed resolved most of my doubts and provided more comprehensive support for the arguments in this paper. I am inclined to accept this paper, but since I initially gave it a score of 3, which means leaning towards acceptance, I will maintain the score of 3.

**Claims And Evidence:**

The claims in the paper are generally well-supported by evidence. The authors clearly demonstrate that existing quantization methods perform poorly on MoE models, showing substantial performance degradation compared to full precision models. The paper provides compelling evidence that their proposed MoEQuant framework improves quantization performance across multiple MoE models (Qwen-MoE-14B, DeepSeek-MoE-16B, Mixtral-8x7B) and evaluation tasks. The performance improvements are substantial - showing gains of more than 10 points on HumanEval for DeepSeek-MoE-16B under 4-bit quantization. The ablation studies effectively demonstrate the individual contributions of EBSS and AGQ, showing that their combination yields the best results. The efficiency claims (3.2× memory savings and 1.2× inference speedup) are supported by benchmark data. **I did not find any claims without evidence**.

**Essential References Not Discussed:**

I think the paper adequately covers the essential references needed to understand the context for its key contributions. But I am not very familiar with this domain.

**Experimental Designs Or Analyses:**

While the overall setup is well-crafted and appropriate, there are several areas that could benefit from further analysis, clarification, or improvement:
- The authors present detailed quantitative results (Tables 1, 2, and 9), but certain observed phenomena are not thoroughly explained. For example: i. Why do some tasks (like HumanEval) benefit significantly from MoEQuant, while others show minimal or no improvement?
ii. Why does AGQ alone underperform GPTQ on certain tasks, but perform better when combined with EBSS?
- Severe accuracy drops occur for some methods at 3-bit quantization (e.g., RTN). However, the authors do not explicitly analyze or discuss the reasons behind these dramatic performance degradations, nor clarify how MoEQuant mitigates these drops.
- Temperature parameter τ selection: In the appendix, the authors show results for different τ values, but provide no theoretical justification for why 1.2 is optimal, nor clearly explain the physical meaning of how this parameter affects the sampling process.

**Methods And Evaluation Criteria:**

The proposed methods (EBSS and AGQ) and the evaluation criteria (datasets and tasks) introduced in the paper are reasonable, relevant, and well-suited to the specific challenges faced when quantizing Mixture-of-Experts (MoE) large language models. However, upon careful review, there is an aspect where the paper could be improved or clarified:
- The paper supports its methods with comprehensive ablation studies that analyze the impact of key hyperparameters (e.g., temperature and branch number in EBSS) and comparisons across multiple models and bitwidth settings. This thorough experimental validation reinforces that the proposed methods are well-tailored to address the quantization challenges in MoE architectures.

**Other Comments Or Suggestions:**

The paper is overall well-prepared, but here are a few additional suggestions and comments:

- Further explain how the temperature parameter (τ) directly affects the sampling process in EBSS, and consider providing more intuition behind the choice of a τ value of 1.2.
- Some experimental results, such as the variations observed in different tasks and the severe drops in accuracy for certain methods at low-bit quantization, could benefit from deeper analysis or discussion.

**Other Strengths And Weaknesses:**

The work robustly tackles calibration imbalances and reduces quantization errors, achieving near full-precision performance even under low-bit quantization. Extensive experiments demonstrate significant gains in accuracy, memory savings, and inference speed, while the integration of theoretical insights with practical validations underscores the overall significance and real-world applicability of the approach. But it also has some weaknesses:

- The derivations assume quasi-linearity in expert network activations (e.g., with ReLU). Although this is reasonable in many cases, it might not hold for all activation functions, potentially limiting the generality of the theoretical results.
- There are certain discrepancies in the experimental results that require further clarification:
As I said in the **Experimental Designs Or Analyses**, the paper does not fully explain why some tasks benefit more significantly from MoEQuant than others.

**Questions For Authors:**

- The experiments indicate that MoEQuant leads to significant improvements on certain tasks while showing minimal gains on others. Can you elaborate on the underlying reasons for these variations? For example, are there characteristics of specific tasks that better exploit the strengths of MoEQuant?

- The paper notes dramatic accuracy drops with methods like RTN at 3-bit quantization. Could you discuss the reasons behind these severe drops and explain how MoEQuant effectively mitigates these issues?

- Could you provide additional theoretical or intuitive insights into how the temperature parameter (τ) affects the sampling process in EBSS? Specifically, how does varying τ influence the balance of calibration samples across experts, and why was a value of 1.2 deemed optimal?

- Your theoretical derivations assume quasi-linearity in expert network activations (e.g., with ReLU). How might the quantization performance be affected if the activations deviate from this quasi-linear behavior (e.g., when using non-linear activations)?

**Relation To Broader Scientific Literature:**

The paper’s contributions build directly upon and extend key ideas from both the quantization literature and prior research on MoE models. It leverages established methods such as GPTQ and AWQ—originally designed for traditional, dense LLMs—while addressing unique challenges inherent to MoE architectures, such as inter- and intra-expert imbalances. The proposed EBSS innovates on earlier load-balancing strategies by ensuring that calibration data more evenly represents the diverse set of experts in an MoE model, a concern that has been highlighted in previous MoE studies. Meanwhile, AGQ draws on and refines existing error-compensation techniques by incorporating token-expert affinity information, which is critical for accurately adapting quantization to the dynamic routing nature of MoE layers. Thus, by integrating ideas from both expert balancing and sensitivity analysis, the work not only situates itself within the broader context of quantization and model compression but also advances these fields by providing tailored solutions for efficient and accurate quantization of MoE-based LLMs

**Theoretical Claims:**

I examined the derivations supporting the affinity-aware quantization error and the modified Hessian computation. These proofs extend standard PTQ formulations by incorporating token-expert affinity—where gating coefficients scale individual token contributions—into the error and sensitivity assessments. Overall, the algebraic steps appear sound under the assumption that the feedforward operations in the expert networks are quasi-linear (a reasonable approximation for activations like ReLU); however, this assumption might not hold for all activation functions, potentially limiting the generality of the results. No explicit calculation errors were found, though the reliance on these approximations suggests that further empirical validation is warranted to ensure that any deviations from quasi-linearity do not adversely impact the quantization performance.

---

> ### Author Rebuttal · Authors · 2025-03-31
>
> We sincerely appreciate your constructive feedback. Below, we provide point-by-point responses, with all revisions incorporated accordingly.
>
> ---
> >Why do some tasks (like HumanEval) benefit significantly from MoEQuant, while others show minimal or no improvement?
>
> Thanks for your insightful observation. We classify tasks into:
> * **Simple  Retrieval Tasks** (e.g., BOOLQ HellaSwag) rely on direct answer based on surface-level knowledge and exhibit limited benefits from multi-expert collaboration. **However, we still reduces the FP accuracy gap by up to 54%(e.g., on DeepSeek-MoE-16B).**
> * **Challenging Resoning Tasks** (e.g., HumanEval GSM8K) involve continuous multi-step logical/mathematical reasoning, code generation, or structured problem-solving. MoE architectures excel here by assigning specialized experts to sub-problems, enhancing performance through collaborative reasoning.
> However, Applying PTQ to MoE LLMs for these tasks introduces two key challenges:
>     * Each expert must be sufficiently calibrated to ensure both functional accuracy and diversity.
>     * Mitigating error accumulation during multi-step decoding, which severely degrades accuracy.
>
>     **MoEQuant addresses them via EBSS(Expert-Balanced Self-Sampling) for comprehensive calibration of each expert and AGQ(Affinity-Guided Quantizatin) for improved quantization error modeling, achieving 5–24% relative accuracy gains.**
> >Why does AGQ alone underperform GPTQ on certain tasks, but perform better when combined with EBSS?
>
> AGQ optimizes quantization error via token-expert affinity weighting. However, **when used alone, imbalanced sample distribution among experts leads experts to receive fewer low-confidence samples,** increasing noise and variance in Hessian and causes evaluation variance. Despite this, AGQ still improves overall performance(1.2% relative gain on Mixtral).
>
> **Combining AGQ with EBSS ensures sufficient calibration for each expert, yielding a more stable Hessian with low variance**, enabling 4% relative gain on Mixtral. This synergy outperforms either method alone.
> > The paper notes dramatic accuracy drops with methods like RTN at 3-bit quantization. Could you discuss the reasons behind it and explain how MoEQuant effectively mitigates these issues?
>
> As shown in [Table R1](https://anonymous.4open.science/r/MoEQuant-DDFD/Tables_for_R1_eTCn.md), 3-bit quantization, limited to just 8 discrete values, constrains RTN performance. **Moreover, MoE LLMs suffer from inter- and intra-expert imbalances, exacerbating degradation.**
> MoEQuant addresses these imbalances via:
> 1. EBSS: Enables comprehensive calibration of each expert, mitigating inter-expert imbalance.
> 2. AGQ: Incorporates token-expert affinities into quantization loss, correcting intra-expert imbalance.
>
> **This dual approach in MoEQuant delivers SOTA 3-bit PTQ results**, yielding up to a 10.5% accuracy gain on Mixtral-8x7B (Table 3).
> >Could you provide additional theoretical or intuitive insights into how the temperature parameter (τ) affects the sampling process in EBSS?
>
> EBSS optimizes the objective:  $$\mathcal{D}^* = \arg\min_{\mathcal{D}} \{
> \overbrace{\text{PPL}(\mathcal{M},\mathcal{D})}^{\in[0,1]} + \overbrace{\frac{\sigma(\mathcal{M},\mathcal{D})}{\tau}}^{\in[1,+\infty)}
> \}$$where $\tau$ balances distribution alignment (via PPL) and expert balance (via $\sigma$). **Smaller τ prioritizes expert balance but may slightly increase PPL. However, PPL remains bounded, as proved below:**
> - When ignoring $\sigma$，we denote $\text{PPL}=p$, $p$ typically remains near 1 due to self-sampling(see Figure 4). We derive:$$\mathcal{D}^* = \text{PPL}(\mathcal{M,D})+ \frac{\sigma(\mathcal{M},\mathcal{D})}{\tau}\leq p+\frac{1}{\tau}$$ since $\sigma>=0$, we derive: $$\text{PPL}(\mathcal{M,D})\leq p+\frac{1}{\tau}$$
>
> The hyperparameter $\tau=1.2$ is determined via ablation studies. Further experiments in [Table R2](https://anonymous.4open.science/r/MoEQuant-DDFD/Tables_for_R1_eTCn.md) confirm its generalizability.
> > How might the quantization performance be affected if the activations deviate from this quasi-linear behavior?
>
> For nonlinear function SiLU used in MoE LLMs, Equation 17 and AGQ remain effective. We support it by corresponding experiments：
> - **Numerical Error.** As shown in [Table R3](https://anonymous.4open.science/r/MoEQuant-DDFD/Tables_for_R1_eTCn.md), the reformulated gate output: $$y_i\approx\left((xW^{up})\odot f(c_ixW^{gate})\right)W^{down}$$ remains high cosine similarity with true outputs across c values, approaching 1 for high-c experts . Top-k expert selection inherently prioritizes high-affinity tokens, further minimizing cumulative error.
> - **Task Performance Validation.** As Shown in [Table R4](https://anonymous.4open.science/r/MoEQuant-DDFD/Tables_for_R1_eTCn.md), applying AGQ to the gate layers (which governs non-linear interactions) yields significant gains in both perplexity and downstream accuracy(1.2% gain for Qwen-MoE), confirming its effectiveness.

---

### Decision · Program_Chairs · 2025-05-01

**Decision:**

Accept (poster)

**Comment:**

This paper proposes new MoE quantization methods called Expert-Balanced Self-Sampling (EBSS) and Affinity-Guided Quantization (AGQ) to address imbalance problem of MoE quantization. EBSS provides a better calibration dataset selection to balance the expert usage, and AGQ uses affinities between experts and samples into the quantization process.

The reviewers generally agreed on the effectiveness of the proposed methods supported by the presented evidence. Therefore, all the reviewers agreed to accept the paper in the conference (1 accept and 3 weak accepts). Reviewers acknowledge the importance of the contributions - "MoEQuant introduces two innovative methods", "The paper is well illustrated, and the method design is clearly stated. The paper studies a timely problem" together with the strong empirical results.

Some reviewers questioned about the task sensitivity, hyper-parameter sensitivity and more detailed comparisons, and the authors provided a thorough investigation to address the issues. The reviewers explicitly and implicitly acknowledged that most of the questions were resolved by the rebuttal.

Given the consensus among reviewers, the paper provides an important contribution to the community, and it is recommended to be appeared in ICML 2025.